# Glutamate spillover in *C. elegans* triggers repetitive behavior through presynaptic activation of MGL-2/mGluR5

Menachem Katz[1], Francis Corson[2,5], Wolfgang Keil[1,3,5], Anupriya Singhal[1], Andrea Bae[1], Yun Lu[1], Yupu Liang[4] & Shai Shaham [1]

Glutamate is a major excitatory neurotransmitter, and impaired glutamate clearance following synaptic release promotes spillover, inducing extra-synaptic signaling. The effects of glutamate spillover on animal behavior and its neural correlates are poorly understood. We developed a glutamate spillover model in *Caenorhabditis elegans* by inactivating the conserved glial glutamate transporter GLT-1. GLT-1 loss drives aberrant repetitive locomotory reversal behavior through uncontrolled oscillatory release of glutamate onto AVA, a major interneuron governing reversals. Repetitive glutamate release and reversal behavior require the glutamate receptor MGL-2/mGluR5, expressed in RIM and other interneurons presynaptic to AVA. *mgl-2* loss blocks oscillations and repetitive behavior; while RIM activation is sufficient to induce repetitive reversals in *glt-1* mutants. Repetitive AVA firing and reversals require EGL-30/Gαq, an mGluR5 effector. Our studies reveal that cyclic autocrine presynaptic activation drives repetitive reversals following glutamate spillover. That mammalian GLT1 and mGluR5 are implicated in pathological motor repetition suggests a common mechanism controlling repetitive behaviors.

[1] Laboratory of Developmental Genetics, The Rockefeller University, 1230 York Avenue, New York, NY 10065, USA. [2] Laboratoire de Physique Statistique, Ecole Normale Supérieure, CNRS, Université Pierre et Marie Curie, Université Paris Diderot, 75005 Paris, France. [3] Center for Studies in Physics and Biology, The Rockefeller University, 1230 York Avenue, New York, NY 10065, USA. [4] Research Bioinformatics, The Rockefeller University, 1230 York Avenue, New York, NY 10065, USA. [5] These authors contributed equally: Francis Corson, Wolfgang Keil. Correspondence and requests for materials should be addressed to S.S. (email: shaham@rockefeller.edu)

Glutamate spillover following synaptic release has been extensively studied as a mechanism for engaging non-synaptic glutamate receptors[1,2]. Such spillover can have physiological roles, such as feed-forward disinhibition of GABAergic interneurons[3]; and pathological consequences, including a suggested role in drug-seeking behavior[4]. Spillover is often studied by inducing high levels of extracellular glutamate, conditions that can also drive neurotoxicity[5,6]. Thus, teasing apart whether consequences of spillover are due to toxicity or to extra-synaptic signaling effects on neural circuits is challenging, particularly in the context of long-term behavioral studies.

The *Caenorhabditis elegans* nervous system is composed of only 302 neurons[7], and viability of these cells is not affected by excess glutamate[8]. Thus, this animal provides an excellent arena in which to investigate specific circuit and behavioral consequences of glutamate spillover, independent of its effects on neuronal health. In vertebrates, astrocytes are major regulators of glutamate homeostasis in the central nervous system. These glial cells are thought to regulate extracellular glutamate levels through active uptake of glutamate using the glial glutamate transporters GLAST/EAAT1 and GLT1/EAAT2[1,5,9]; and passively, by ensheathing, and thereby physically insulating synapses[10]. The four *C. elegans* CEPsh glia share developmental, physiological, and functional properties with mammalian astrocytes[11–15], and like astrocytes, tightly associate with the central brain neuropil housing most synapses in the animal (the nerve ring[7]; Fig. 1a).

To establish a glutamate spillover paradigm in *C. elegans* that might be broadly informative, we aimed to determine whether the CEPsh glia also control glutamate levels; whether this control is exerted through active glutamate uptake, as in astrocytes[1,5,9]; and, if so, whether it has an effect on synaptic activities. We used transcriptome profiling to demonstrate molecular similarities between *C. elegans* CEPsh glia and murine astrocytes, and found that mRNAs for *glt-1*, encoding a homolog of GLT1/EAAT2, are enriched in these glia. Postembryonic ablation of CEPsh glia, or GLT-1 loss, promotes two different glutamate-dependent behavioral defects: an increased overall rate of locomotory reversal events, and a surprising clustering of reversals in repetitive bouts. In vivo measurements of extracellular glutamate dynamics near postsynaptic sites of the interneuron AVA, a major regulator of reversal behavior, show that GLT-1 is required to prevent oscillations in synaptic glutamate release, and consequent oscillations in AVA activity, which correlate with repetitive reversal events in frequency. Furthermore, both oscillations in AVA neuron activity and repetitive behavior are mediated by activation of the metabotropic glutamate receptor MGL-2/mGluR5 in glutamatergic interneurons presynaptic to AVA. Aberrant repetitive AVA activity and reversal behavior require the excitatory *egl-30*/Gαq signaling pathway.

Our studies suggest a simple, yet general, mechanism by which aberrant glutamate spillover can induce repetitive behavior through cyclic autocrine presynaptic activation. Our findings also

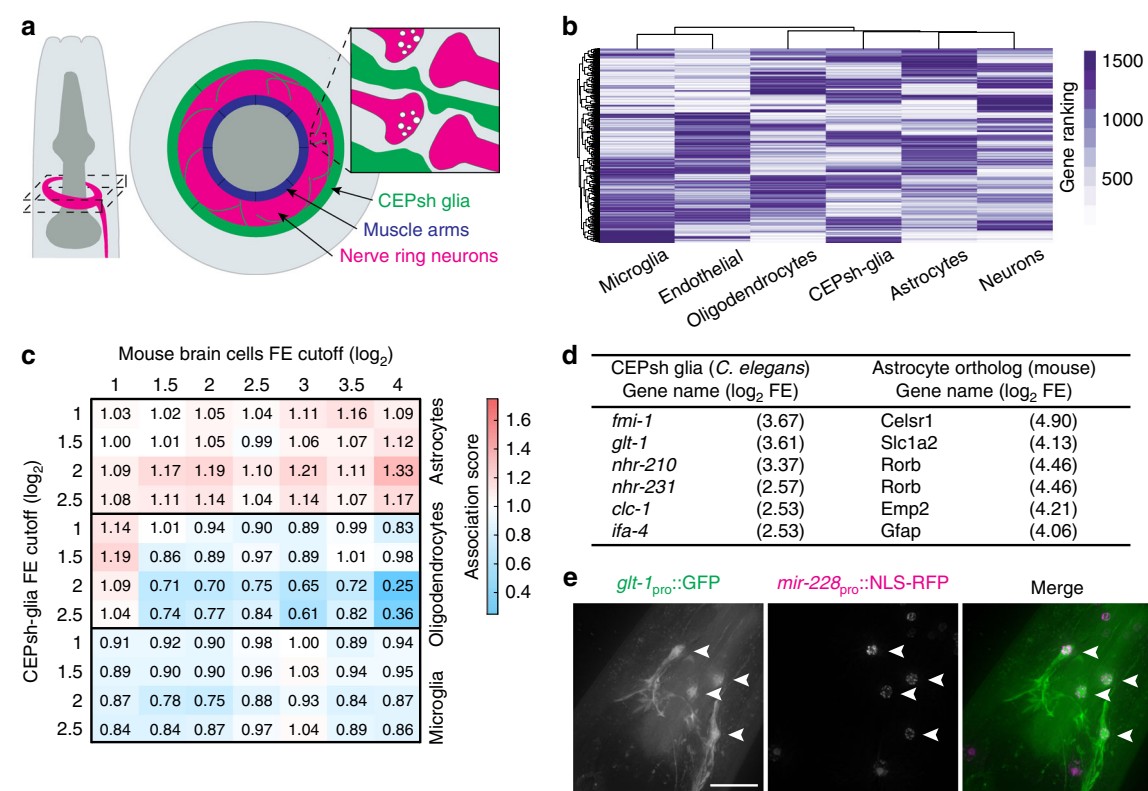

**Fig. 1** CEPsh glia share molecular similarities with astrocytes, including GLT-1 expression. **a** Schematic of *C. elegans* head (left), anterior is up, ventral is right. Cross-section of dashed area is enlarged (middle). Magenta, nerve ring; green, CEPsh glia; blue, muscle arms. Interactions of CEPsh glia processes with synapses are illustrated (inset). **b** Hierarchical clustering of mouse brain cells and CEPsh glia-expressed genes. *x*-axis dendrogram indicates the relationships among different cell types, *y*-axis dendrogram indicates relationships across individual genes. **c** Overrepresentation comparison of CEPsh glia-enriched genes with gene orthologs enriched in mouse glial cells. Normalized association values indicated by numbers and heatmap (FE, fold-enrichment). **d** Examples of CEPsh glia-enriched genes with orthologous genes enriched in mouse astrocytes. **e** Maximum intensity projections of deconvolved optical sections. *glt-1* promoter expression in CEPsh glia (left; arrowheads), *mir-228* glial promoter expression (middle; CEPsh glia, arrowheads); merged image (right). Scale bar, 8 μm

suggest a plausible circuit origin for repetitive behavior in the context of human neuropsychiatric illness.

## Results

**CEPsh glia share molecular similarities with astrocytes.** To establish a glutamate spillover paradigm in *C. elegans*, we first sought to examine molecular similarities between CEPsh glia and astrocytes, particularly in the context of glutamate homeostasis. We therefore sequenced CEPsh glia mRNAs, and assessed their abundance and enrichment relative to RNA levels of all other *C. elegans* cells. We then compared the list of CEPsh-enriched genes with lists of mouse homologs enriched in different brain cell types[16] using hierarchical clustering. We found that CEPsh glia cluster together with mouse neurons, astrocytes and oligodendrocytes, and separately from microglia and endothelial cells (Fig. 1b). This is in line with the developmental origins of these cells: *C. elegans* CEPsh glia, and mammalian neurons, astrocytes, and oligodendrocytes, are all ectodermally derived, whereas microglia and endothelial cells are of mesodermal origin[16]. Rank-ordering mRNAs, according to fold-enrichment (FE) within a cell type, reveals that the more highly enriched a gene is in CEPsh glia, the more likely is its ortholog to be enriched in murine astrocytes, but not in oligodendrocytes (or microglia; Fig. 1c, Supplementary Data 1). For example, some genes enriched in both cell types include *ifa-4*, homologous to the glial fibrillary acidic protein (GFAP), known to affect astrocyte morphology and activity[17]; *fmi-1*, homologous to Flamingo (Celsr1), a regulator of axon migration and fasciculation[12]; and *glt-1*[18], homologous to the astrocyte-expressed Slc1a2/GLT1/EAAT2 glutamate transporter[5,6] (Fig. 1d, Supplementary Data 1). Thus, of the different mouse brain cells in our comparisons, CEPsh glia are most similar to astrocytes by molecular, morphological[7,12], and functional[12,13,15] criteria.

Since GLT1 is a known astrocyte regulator of glutamate homeostasis[5], we further characterized *glt-1* function in CEPsh glia. To confirm *glt-1* gene expression in CEPsh glia, we generated animals carrying a transgene in which *glt-1* upstream regulatory sequences drive expression of GFP. Strong GFP expression is observed in all four CEPsh glia (Fig. 1e). Weaker expression is also detected in muscles, consistent with a previous report[18].

**CEPsh glia-expressed GLT-1 regulates reversal behavior.** Decreasing or increasing glutamate signaling in *C. elegans* using mutations blocking presynaptic glutamate release (e.g. in *eat-4*/VGLUT[19]), or mutations activating glutamate receptors[20], respectively, is known to decrease/increase locomotory reversal rates, respectively. Thus, to study glial *glt-1* regulation of glutamate signaling, we recorded the locomotion of animals exploring their environment, a behavior characterized by infrequent reversals[21]. By analyzing these movies using a Hidden Markov Model with data-derived transition probabilities between forward and backward locomotion states (Supplementary Fig. 1a), we found that postembryonic ablation of CEPsh glia (which does not affect neuronal viability or nerve ring architecture; Supplementary Fig. 2a–c), or a mutation in *glt-1*, aberrantly increases animal reversal rates to similar extents (Fig. 2a and Supplementary Fig. 1b–e; consistent with previous observations[18]). These excessive reversals are attenuated in animals also carrying a mutation in the vesicular glutamate transporter *eat-4*/VGLUT (Fig. 2a). This is consistent with the notion that excessive reversals are dependent on presynaptic glutamate release. While CEPsh glia ablation induces additional locomotion defects[15] (Supplementary Movies 1–2), *glt-1* loss does not (Supplementary Movie 3). Thus, *glt-1* modulates glutamate-dependent reversal

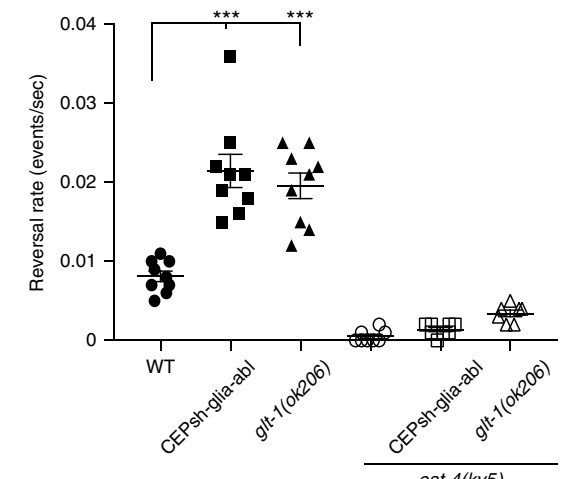

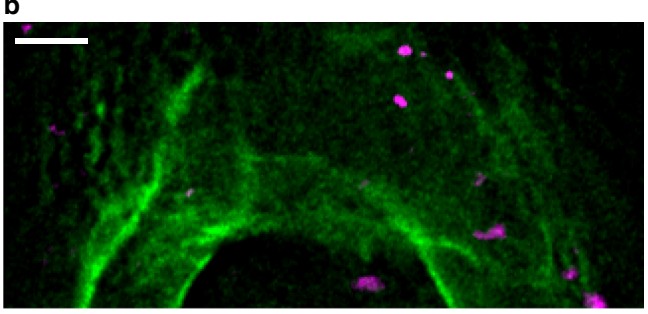

glr-1_pro::NLG-1::mCherry
hlh-17_pro::GLT-1::GFP

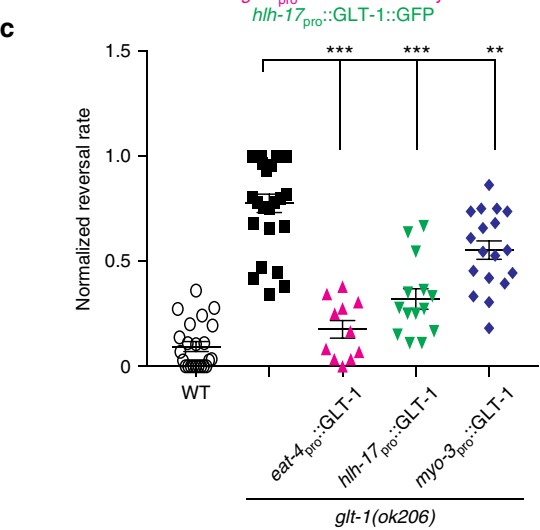

**Fig. 2** *C. elegans* reversal behavior is controlled by CEPsh glia-expressed *glt-1*. **a** Reversal rates in wild type (WT, black circles; n = 9 movies), CEPsh-glia-ablated (black boxes; n = 9 movies), *glt-1(ok206)* (black triangles; n = 9 movies), *eat-4(ky5)* (open circles; n = 8 movies), *eat-4(ky5)*; CEPsh-glia-ablated (open boxes; n = 8 movies) and *eat-4(ky5)*; *glt-1(ok206)* (open triangles; n = 7 movies) strains. **b** A deconvolved optical section. GLT-1 fused to GFP expressed in CEPsh glia using *hlh-17* promoter. Synapses are marked by glr-1_pro::NLG-1::mCherry. Scale bar, 4 μm. **c** Cell-specific *glt-1(ok206)* rescue studies. Reversal rates normalized relative to minimum value of WT controls and maximum value of *glt-1* mutants. Wild type (WT, open circles; n = 23 movies), *glt-1(ok206)* (black boxes; n = 25 movies), GLT-1 transgene expressed in neurons (magenta triangles; *eat-4*; n = 11), CEPsh glia (*hlh-17*, green triangles; n = 14), muscles (*myo-3*, blue diamonds; n = 18). **a, c** Bars, mean ± SEM, ANOVA Tukey's HSD post hoc test, **p < 0.005, ***p < 0.0005. Source data are provided as a Source Data file

behavior[18], and lack of glia-expressed GLT-1 is likely the primary cause of reversal behavior defects in CEPsh glia-ablated animals.

Consistent with the idea that *glt-1* influences synaptic glutamate signaling, we found that most glutamatergic synapses, marked by mCherry-tagged neuroligin (NLG-1) expressed in ionotropic glutamate receptor (*glr-1*)-positive neurons, are within 0.5 μm of glial processes expressing GLT-1::GFP (Fig. 2b, Supplementary Fig. 3 and Movie 4). To determine the functional relevance of this synaptic proximity, we assessed the rescue of *glt-1* mutant behavioral defects in animals expressing GLT-1 protein specifically in either CEPsh glia (within 0.5 μm of synapses; using the *hlh-17* promoter element), or in muscles (within 2 μm of synapses; using the *myo-3* promoter element), whose arms synapse to motor neurons at the inner facet of the nerve ring, but do not penetrate the neuropil[7]. We found that glial expression of GLT-1 restores normal reversal rates more effectively than muscle-specific expression (Fig. 2c). Further supporting a proximity model, forcing GLT-1 expression directly at synapses, by ectopic expression in glutamatergic neurons (using the *eat-4* promoter element), where *glt-1* is normally not endogenously expressed, results in the most efficient rescue (Fig. 2c).

In summary, our data suggest that loss of *glt-1* provides an appropriate setting for studying the effects of increased extracellular glutamate, allowing us to interrogate effects of glutamate spillover on circuit function and behavior.

**GLT-1 functions to prevent repetitive reversal behavior**. To examine the behavioral consequences of *glt-1* loss in detail, we examined the temporal distribution of spontaneous reversal events of *glt-1*(−) animals moving on agar plates. Specifically, we analyzed the probability, $p_f(t)$, that an animal performing a reversal event will not initiate a reversal over a subsequent period of time, $t$. In wild-type animals, $p_f(t)$ is well fit by a decaying single exponential function, consistent with a random Poisson process in which the probability of reversal per unit time is fixed (Fig. 3a). In *glt-1*(*ok206*) mutants, we find two deviations from the wild-type reversal pattern. First, because *glt-1* mutants have a higher average reversal rate (Fig. 2a), an exponential fit based on this average rate has a steeper slope than seen in the wild type (black line, compare Fig. 3a, b). Second, and more important, the experimentally observed *glt-1*(*ok206*) reversal distribution is not well fit by a single exponential function (compare red and black lines in Fig. 3b). In these mutants, the probability of reversal within 25−30 s following a reversal event is high, and the proportion of long reversal-free intervals following an initial reversal is reduced (Fig. 3b). Indeed, modeling *glt-1* mutant reversals with a fixed reversal rate, boosted by a perturbation at short times after a reversal, fits our data well (Supplementary Fig. 4a, b; see Methods). The weight of the perturbation defines a natural repetitive behavior index (RI), which we use to quantify the deviation from the Poisson process (Fig. 3d and Supplementary Fig. 4a, b; see Methods). This defines a general approach to

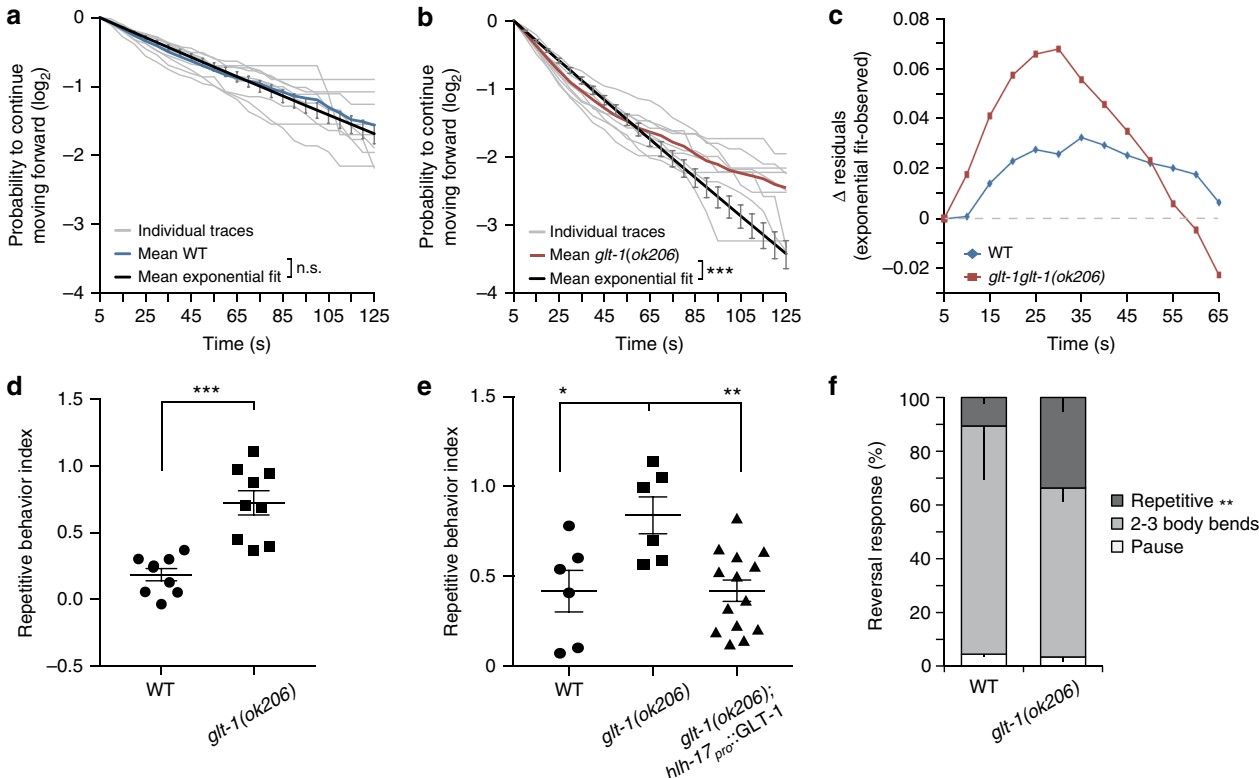

**Fig. 3** GLT-1 prevents repetitive reversal behavior. **a** Probability that following a reversal, a wild-type animal will move forward for the time indicated on the horizontal axis. Exponential fits are plotted in black ± SEM, light gray lines are probability distributions of individual movies (blue, mean plot; $n = 9$ movies). **b** Same as (**a**), except in *glt-1* mutants (red, mean plot; $n = 9$ movies). **a**, **b** $\chi^2$-test used to determine if exponential fit differs from mean observed data (colored lines; n.s., nonsignificant; ***$p < 10^{-15}$). **c** The residual difference between the mean exponential fit and the mean observed reversal probability for wild-type (WT; blue) and *glt-1* (red) mutants at the indicated time intervals. **d** *glt-1* mutants exhibit repetitive reversal behavior (from data in (**a**) (circles), (**b**) (boxes), Student's *t* test, ***$p < 0.0005$. **e** GLT-1 in CEPsh glia rescues repetitive reversals (ANOVA Tukey's HSD post hoc test, *$p < 0.05$, **$p < 0.005$; WT (circles) $n = 6$, *glt-1*(*ok206*)(boxes) $n = 6$, *glt-1*(*ok206*); *hlh-17*<sub>pro</sub>::GLT-1 (triangles) $n = 14$ movies. **f** *glt-1* mutants respond to mechanical stimuli with repetitive reversals (Student's *t* test, **$p < 0.005$; WT $n = 16$, *glt-1*(*ok206*) $n = 13$ animals). **d**–**f** Bars, mean ± SEM. Source data are provided as a Source Data file

quantitatively measure what has been anecdotally termed repetitive behavior in other settings. Confirming the importance of glia in glutamate clearance, expression of GLT-1 in CEPsh glia inhibits repetitive reversals in *glt-1(ok206)* mutants (Fig. 3e).

To investigate if GLT-1 is a general inhibitor of repetitive reversals, we examined reversals associated with the escape response to aversive stimuli. Anterior mechanical stimuli applied to wild-type *C. elegans* induce backward movement for ~3 body bends, followed by forward movement in a different heading[22]. Similar stimulations of *glt-1* mutants result in more consecutive reversals than in wild-type animals (Fig. 3f, Supplementary Movies 5,6). Thus, repetitive reversals in *glt-1(ok206)* mutants are induced in two distinct behavioral contexts (exploration and escape), demonstrating that GLT-1 functions generally to prevent repetitive reversal behavior.

**GLT-1 blocks oscillatory glutamate release and AVA activity.** To uncover the underlying circuit disruptions leading to repetitive behavior in *glt-1* mutants, we focused on AVA, a major *C. elegans* interneuron driving backward locomotion[22,23]. AVA expresses several glutamate receptors[24], and increased extracellular glutamate near AVA postsynaptic sites correlates with

AVA calcium responses[25] that induce reversal[25,26]. We recorded spontaneous AVA activity in transgenic animals expressing both the fluorescent glutamate sensor iGluSnFR[25] and the calcium indicator RCaMP. Animals were trapped in a microfluidic chamber, paralyzed using a muscle cholinergic receptor antagonist (tetramisole), and continually exposed to food supernatant, which inhibits reversal behavior[21], and allows synchronization of our recordings. Wild-type animals initially show no glutamate release or calcium induction events in our assay. However, after 4 min ± 24 s (mean ± SEM) of habituation to the food cue, spontaneous glutamate release events onto AVA ensue, and these are highly correlated with AVA calcium transients (average peak cross-correlation 0.8; Fig. 4a and Supplementary Fig. 5a, Supplementary Movie 7). *glt-1* mutants habituate more rapidly than wild-type animals (2 min ± 22 s on average), as might be expected for animals that have a higher propensity for activating glutamatergic signaling. Remarkably, *glt-1* mutants often exhibit correlated long-duration oscillations of glutamate and AVA calcium signals following food cue habituation, suggesting repeated activation of AVA (Fig. 4b and Supplementary Fig. 5b, Supplementary Movie 8; average peak cross-correlation 0.7).

AVA calcium and glutamate signals are highly dynamic, with transient oscillations of varying frequencies and intensities, but

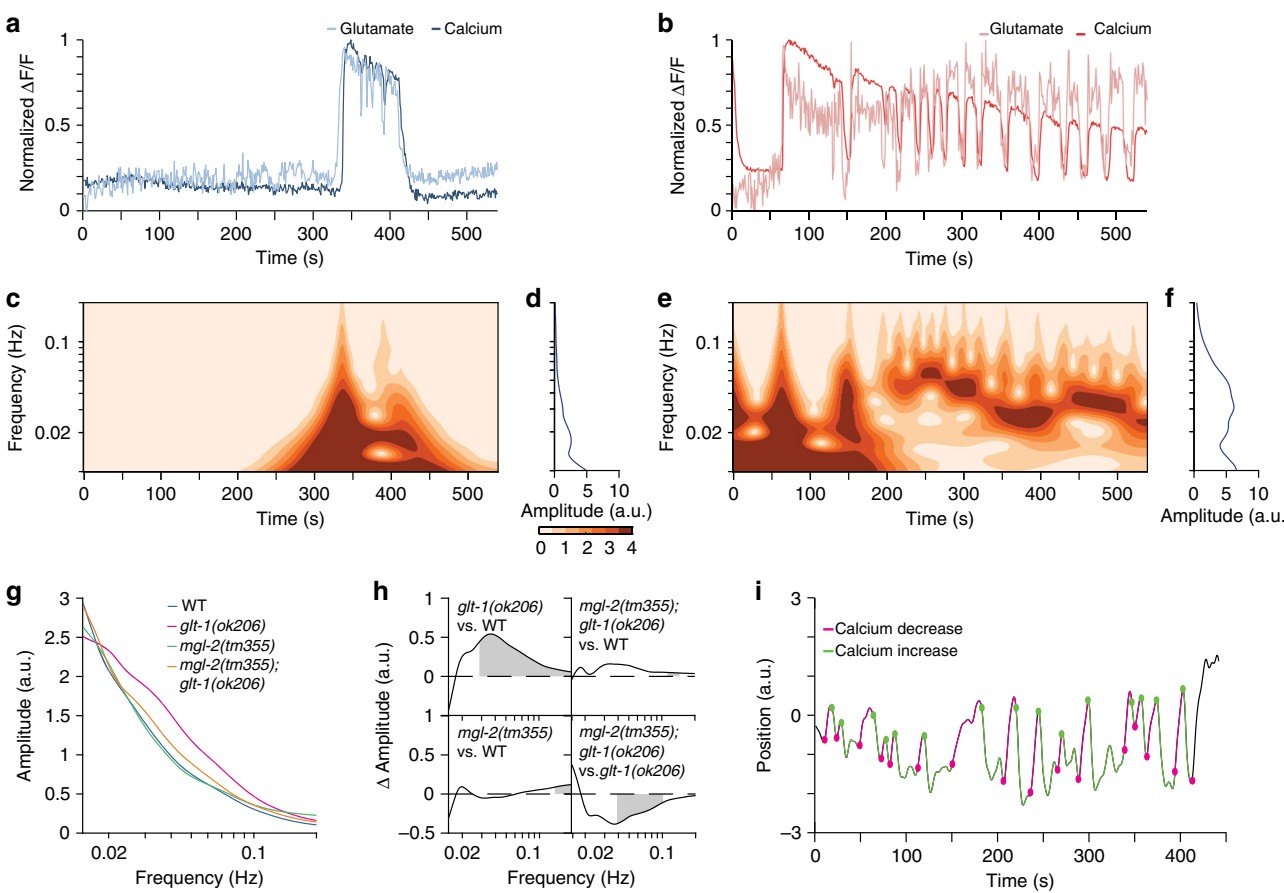

**Fig. 4** GLT-1 inhibits MGL-2-dependent oscillations in glutamate secretion and AVA activation. **a** Representative traces of spontaneous glutamate (light blue) and calcium (dark blue) dynamics near and in a wild-type AVA neuron, respectively. **b** As in (**a**), *glt-1(ok206)* mutant (glutamate, light red; calcium, dark red). **c** Time-dependent frequency spectrum of calcium traces of the wild-type animal in (**a**). **d** Time-averaged frequency amplitudes from the wavelet spectrum in (**c**). **e**, **f** As in (**c**, **d**), *glt-1(ok206)* mutant. **g** Mean time-averaged frequency amplitudes for WT (blue line; 20 traces), *glt-1(ok206)* (magenta line; 22 traces), *mgl-2(tm355)* (green line; 15 traces) and *glt-1(ok206); mgl-2(tm355)* (orange line; 17 traces). **h** Frequency amplitude differences between indicated strain pairs. Shading, areas of significant difference $p < 0.05$, permutation test from bootstrapped ensembles, $n = 10^4$ bootstrap samples. **i** Representative trace of head motion in nonparalyzed *glt-1* mutant animal between forward (positive) and backward (negative) positions. Traces are decorated in green, magenta for increased, decreased calcium signals, respectively. Circles, transition points. Source data are provided as a Source Data file

exhibit common features among animals of the same genotype. To quantify these common features, we used wavelet analysis to assign local oscillation frequencies at each time point, generating time-dependent frequency spectra (see Methods). This analysis reveals the presence of mostly low frequencies (long duration between peaks) for the calcium responses seen in wild-type animals (<0.02 Hz; Fig. 4a, c, d; Supplementary Fig. 5a, c). The repetitive calcium activities seen in the *glt-1* mutant, however, exhibit pronounced enrichment of oscillations at ~0.04 Hz (Fig. 4b, e, f; Supplementary Fig. 5b, d). This enrichment can be clearly seen in time-averaged frequency amplitudes, averaged over all recorded animals (Fig. 4g, compare magenta and blue curves), demonstrating significant enrichment of oscillations above 0.03 Hz (peak at 0.04 Hz; Fig. 4h, top left). Importantly, these frequencies correspond to calcium induction events peaking at ~25 s intervals, matching the time interval between consecutive reversal events in our behavior assays (Fig. 3c). Supporting this correlation, nonparalyzed animals in our microfluidic setup show strong association between AVA calcium oscillations and attempted transitions between reversal/forward movements[26] (phi coefficient = 0.71; Fig. 4i). Together, these studies are consistent with the idea that glutamate oscillations drive repetitive reversal behavior events in *glt-1* mutants.

**MGL-2 drives oscillatory glutamate release and AVA activity**. To further support this idea, and to elucidate the mechanism of

oscillatory glutamate release onto AVA, we sought mutants abolishing *glt-1*-dependent glutamate release onto and calcium oscillations in AVA. We reasoned that repeated glutamate secretion in *glt-1* mutants could be mediated by extra-synaptic neuronal glutamate receptors responding to glutamate spillover[1]. The enrichment of 0.03−0.1 Hz oscillations (indicating activity peak separation of seconds to tens of seconds) suggest that receptors with slow activation/signaling kinetics, such as metabotropic glutamate receptors (mGluRs)[27,28], may be involved. *C. elegans* encodes three mGluR homologs, MGL-1, MGL-2, and MGL-3, corresponding to mammalian subgroups II, I, and III, respectively[29]. Loss-of-function mutations in *mgl-1*, *-2*, or *-3* do not result in significant changes in glutamate release onto or calcium accumulation in AVA (Fig. 4g, h and Supplementary Fig. 6a, c, d). However, loss of *mgl-2* (but not *mgl-1* or *mgl-3*) significantly suppresses the high-frequency oscillations of *glt-1* mutants (Fig. 4g, h and Supplementary Fig. 6b−d). Thus, glutamate oscillations and subsequent AVA calcium transients require MGL-2 activity. Importantly, loss of *mgl-2* in *glt-1* mutants also significantly, but not completely, ameliorates repetitive reversal behavior (Fig. 5a), supporting the idea that glutamate oscillations drive repetitive reversal bouts. The incomplete reduction in repetitive reversal may indicate that additional glutamate-related mechanisms are involved in this process.

Together, our observations support the notion that repetitive reversals in *glt-1* mutants are caused by oscillations in glutamate release onto AVA.

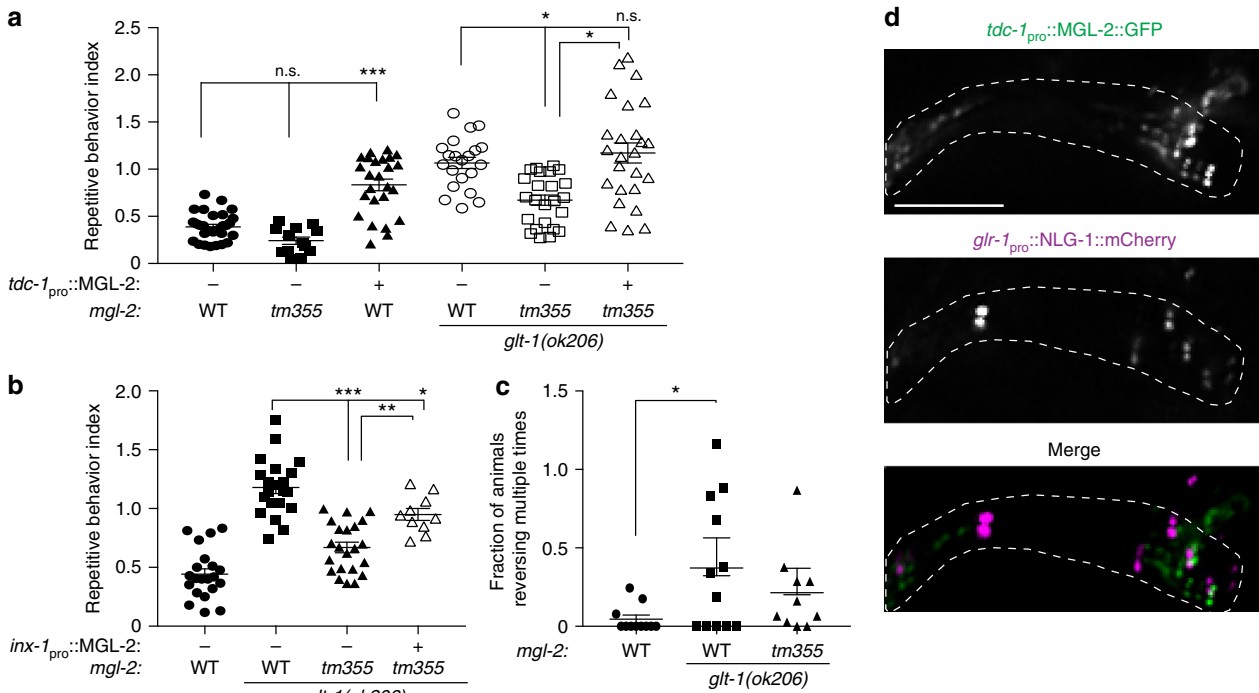

**Fig. 5** Glutamate spillover induces repetitive reversals by presynaptic activation of MGL-2. **a** *mgl-2* mutation ameliorates increased repetitive reversals of *glt-1* mutants. The effect is reversed by re-expression of MGL-2 specifically in RIM neurons (Kruskal−Wallis ANOVA with Dunn's multiple comparison test, n.s., nonsignificant, ***p < 0.0005, *p < 0.05; WT, black circles, n = 30, *mgl-2(tm355)*, black boxes, n = 13, *tdc-1*pro::MGL-2, black triangles, n = 25, *glt-1 (ok206)*, open circles, n = 21, *glt-1(ok206); mgl-2(tm355)*, open boxes, n = 24, *glt-1(ok206); mgl-2(tm355); tdc-1*pro::MGL-2, open triangles, n = 25 movies). **b** MGL-2 expression in AIB neurons rescues repetitive reversals (WT, black circles, n = 21, *glt-1(ok206)*, black boxes, n = 21, *glt-1(ok206); mgl-2(tm355)*, black triangles, n = 22, *glt-1(ok206); mgl-2(tm355); inx-1*pro::MGL-2, open triangles, n = 10 movies). **c** Channelrhodopsin activation of RIM promotes GLT-1-mediated repetitive reversals (WT, black circles, n = 11, *glt-1(ok206)*, black boxes, n = 12, *glt-1(ok206); mgl-2(tm355)*, black triangles, n = 10 movies). **a–c** Bars, mean ± SEM. **b, c** ANOVA Tukey's HSD post hoc test, ***p < 0.0005, *p < 0.05. In (**c**), while not significant at the level of p < 0.05, the trend towards fewer repetitive events in animals also lacking *mgl-2* is consistent with our observations in (**a, b**). **d** GFP-tagged MGL-2 localizes to RIM neurite (left), but not to mCherry-labeled synapses (middle); merge (right). Scale bar, 5 μm; nerve-ring region is indicated with dotted line; 2-μm-thick sum intensity projections. Source data are provided as a Source Data file

**MGL-2 acts extrasynaptically in neurons presynaptic to AVA.** If glutamate spillover drives oscillatory glutamate release and consequent repetitive behavior in *glt-1* mutants, we might expect MGL-2 to function in neurons presynaptic to AVA. Indeed, we found that *mgl-2* is strongly expressed in RIM, a glutamatergic interneuron presynaptic to AVA, and more weakly in AIB, RIA, ADA and AIM, additional glutamatergic presynaptic neurons[7] (Supplementary Fig. 7a and Supplementary Table 1). MGL-2 is not expressed in AVA or in CEPsh glia. Expression of MGL-2 specifically in RIM using a *tdc-1* promoter::*mgl-2* transgene, can significantly increase repetitive behavior in *mgl-2; glt-1* double mutants to a level that is similar to the behavior of *glt-1* mutants alone (Fig. 5a). Similarly, MGL-2 expression in AIB, using an *inx-1* promoter element, also increases repetitive reversals in the double mutant, but to a lesser extent (Fig. 5b). These observations suggest a role for MGL-2 as a general amplifier of repetitive behavior upon glutamate spillover. Ectopic overexpression of MGL-2 in RIM can even induce repetitive reversals in animals with intact *glt-1*, probably due to hyper-sensitization of RIM to lower levels of extracellular glutamate (Fig. 5a). Moreover, MGL-2 expression in RIM restores glutamate and calcium oscillations in *mgl-2; glt-1* double mutants (Supplementary Fig. 7b, compare animals with and without transgene array). Rescue variability in these latter studies is high, likely because of variations in transgene copy number and expression.

We next asked whether RIM activation is sufficient to induce repetitive reversals when glutamate clearance is impaired. In agreement with a previous report[30], light activation of channelrhodopsin-expressing RIM promotes single reversal events in a retinal-dependent manner in wild-type animals (Fig. 5c and Supplementary Fig. 7c). RIM activation in *glt-1* mutants, however, can induce 2−3 short reversal events during light exposure, and these repetitive events appear to be partially reduced by loss of *mgl-2* (Fig. 5c). This observation indicates that spillover of glutamate, autonomously secreted by RIM, can feedback onto RIM to induce repetitive behavior. It also suggests that RIM is unlikely to modulate activity of its presynaptic neurons to induce repetitive behavior.

Finally, since the repetitive behavior of *glt-1* mutants can be elicited by glutamate spillover from RIM output synapses, MGL-2 should localize near but not at these synaptic sites. Indeed, GFP-tagged-MGL-2 exhibits punctate expression in the RIM neurite, adjacent to, but not overlapping glutamatergic synapses (Fig. 5d).

Taken together, these observations suggest that MGL-2 functions extrasynaptically in glutamatergic neurons presynaptic to AVA to mediate repetitive reversals caused by glutamate spillover.

**Repetitive reversal behavior is mediated by Gαq signaling.** How might MGL-2 effect glutamate release? The postsynaptic signaling mechanism of mammalian mGluR5, related to MGL-2, through Gαq has been extensively explored[28]. However, how mGluR5, which localizes to both dendritic and axonal sites[31,32], functions presynaptically is not well understood. We therefore investigated whether Gαq might mediate the presynaptic effects of MGL-2. In *glt-1* mutant animals also defective in *egl-30*/Gαq[33], repetitive reversals are nearly entirely abrogated (Fig. 6a), and a general decrease in reversal rates is observed (Supplementary Fig. 8a). No AVA-related glutamate or calcium oscillations are observed in these double mutants, and even single *egl-30*/Gαq mutants hardly show calcium activity (Supplementary Fig. 8b; hence wavelet analysis cannot be applied for these activity traces). These observations identify EGL-30/Gαq as a required mediator of both reversal initiation and repetitive reversal behavior. To directly examine these roles, we expressed EGL-30 in RIM neurons of

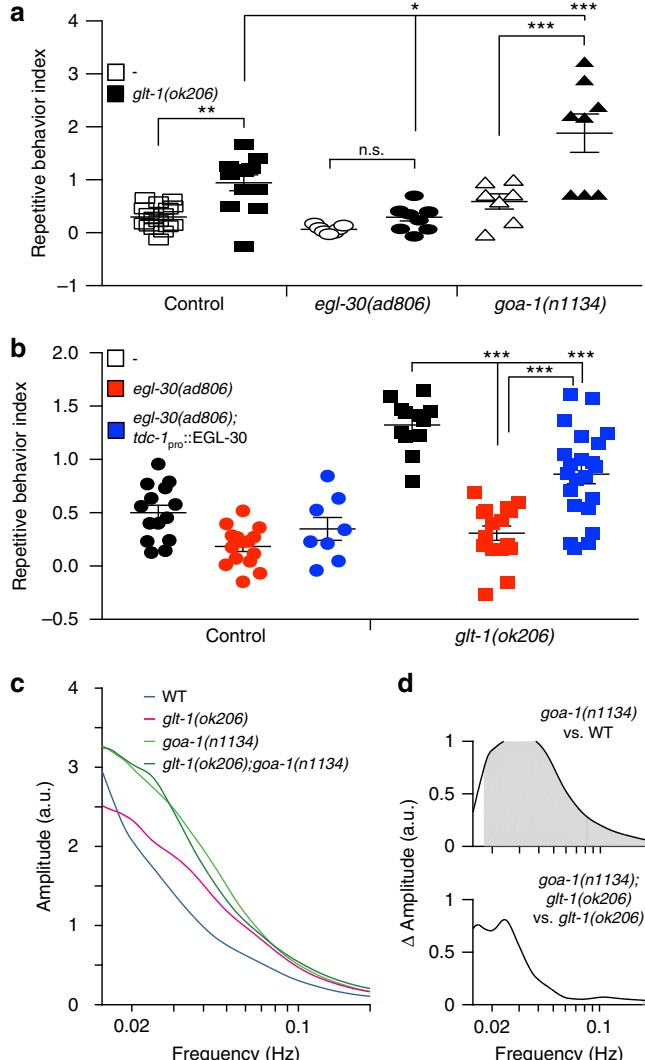

**Fig. 6** Glutamate spillover induces repetitive reversals via the *egl-30*/Gαq signaling pathway. **a** *egl-30*/*goa-1* mutations ameliorate/exacerbate repetitive reversals of *glt-1* mutants (WT, open boxes, $n = 17$, *glt-1(ok206)*, black boxes, $n = 12$, *egl-30(ad806)*, open circles, $n = 7$, *glt-1(ok206); egl-30 (ad806)*, black circles, $n = 10$, *goa-1(n1134)*, open triangles, $n = 7$, *glt-1 (ok206); goa-1(n1134)*, black triangles, $n = 8$ movies). **b** EGL-30 expression in RIM neurons rescues repetitive reversals (WT, black circles, $n = 14$, *egl-30(ad806)*, red circles, $n = 15$, *egl-30(ad806); tdc-1_pro::EGL-30*, blue circles, $n = 8$, *glt-1(ok206)*, black boxes, $n = 12$, *glt-1(ok206); egl-30(ad806)*, red boxes, $n = 16$, *glt-1(ok206); egl-30(ad806); tdc-1_pro::EGL-30*, blue boxes, $n = 22$ movies). **a, b** Bars, mean ± SEM, ANOVA Tukey's HSD post hoc test, n.s., nonsignificant, ***$p < 0.0005$, **$p < 0.005$, *$p < 0.05$. **c** Mean time-averaged frequency amplitudes for WT (blue line, 20 traces), *glt-1(ok206)* (magenta line, 22 traces), *goa-1(n1134)* (light green, 16 traces) and *glt-1 (ok206); goa-1(n1134)* (dark green, 10 traces). **d** Mean time-averaged frequency amplitude differences between indicated strain pairs. Shading, areas of significant differences $p < 0.05$, permutation test from bootstrapped ensembles, $n = 10^4$ bootstrap samples. Source data are provided as a Source Data file

either *egl-30* single mutants or *egl-30*; *glt-1* double mutants. Transgenic *egl-30* single mutants exhibit a low reversal rate similar to nontransgenic animals (Supplementary Fig. 8c), suggesting that *egl-30* expression in RIM is not sufficient to stimulate reversal initiation. However, RIM::EGL-30 expression in *egl-30*; *glt-1* double mutants does significantly increase repetitive

behavior (Fig. 6b). Thus, EGL-30 functions in RIM to induce repetitive behavior, and outside of RIM to promote reversal initiation. Consistent with these observations, the *goa-1*/Gαo pathway in *C. elegans* inhibits the *egl-30*/Gαq pathway[33], and loss-of-function *goa-1(n1134)* mutants exhibit increased reversal rates (Supplementary Fig. 8a), glutamate secretion, and AVA activation events (Fig. 6c, d and Supplementary Fig. 8b), even when *glt-1* is intact. Importantly, a strong increase in repetitive reversals is observed in *goa-1*; *glt-1* double mutants compared to *glt-1* single mutants alone (Fig. 6a), consistent with increased propensity for glutamate release. These findings indicate that repetitive reversals are induced through presynaptic excitations promoted by *egl-30*/Gαq and inhibited by *goa-1*/Gαo.

## Discussion

Taken together, our data suggest a model (Fig. 7) in which conditions permitting glutamate spillover beyond synapses, such as impaired clearance by glial GLT-1, allow glutamate to bind extrasynaptically localized MGL-2/mGluR5 on presynaptic neurons. This induces presynaptic neuron activation, via Gαq, promoting a stimulus-independent glutamate secretion event that leads to firing of the postsynaptic reversal command interneuron AVA. The cycle then repeats, driving repetitive reversal behavior. Termination of repetitive bouts could occur once synaptic vesicle pools are depleted. Consistent with such an autocrine excitatory feedback model, calcium increases in the presynaptic RIM or AIB and postsynaptic AVA neurons are known to be correlated[30].

To generate oscillations, our model predicts that glutamate clearance from the *glt-1(−)* synapses should occur more rapidly than signaling through MGL-2/mGluR5. This idea is plausible, as glutamate has been estimated in other settings to diffuse at millisecond rates ($\sim 0.4\,\mu m^2$/msec[34]), whereas signaling kinetics downstream of metabotropic glutamate receptors have been measured at seconds or tens of seconds[27,28].

While the studies we present here mainly dissect RIM-AVA synapses, the proximity of CEPsh glia membranes to other glutamatergic synapses, coupled with expression of MGL-2 in other glutamatergic neurons, suggests that repetitive reversal behavior could be initiated at other synapses, whose activity is then propagated through the circuit to AVA. A recently described method for global analysis of neuron activation in *C. elegans*[26] could be used to test this idea.

The studies we present here implicate uncontrolled extracellular glutamate levels as a driver of repetitive behavior. In that context, it is interesting that impaired glutamate signaling in humans has also been implicated in the pathophysiology of disorders accompanied by repetitive behaviors[35,36]. Clearance of extracellular glutamate by active transport might be very relevant here. Linkage and copy number variation analyses implicate chromosome regions 11p12–p13 in Autism Spectrum Disorder (ASD)[35], with *SLC1A2* (GLT1/EAAT2) located close to maximal linkage peaks. Consistent with this, WAGR (Wilms tumor, Aniridia, Genitourinary malformations and mental Retardation) syndrome patients show features of autism and have deletions in this transporter gene[37]. Furthermore, mutations in the endosomal cation/proton antiporter NHE9, which promotes GLAST/EAAT1 membrane trafficking in astrocytes, confer susceptibility to ASD[38]. *SLC1A1*(EAAC1), a neuronal glutamate transporter gene, is associated with Obsessive Compulsive Disorder (OCD)[36] and ASD[39], and clinical trials for OCD treatment involve drugs that target glutamate signaling, which might affect glial uptake of glutamate[36].

Murine repetitive grooming is a common model for the study of mechanisms regulating pathological repetitive behavior. Consistent with the repetitive behavior mechanism we describe here, astrocyte-specific knockout of GLT-1 in mice produces repetitive grooming behavior[40]. Additionally, mGluR5 inhibitors rapidly suppress repetitive grooming in mouse models for OCD and ASD[41,42]. A previous connection between GLT-1 and mGluR5, however, had not been noted. Furthermore, the underlying circuitry driving repetitive grooming behavior mediated by modulating these mouse proteins is not understood in detail. That the same proteins act together in repetitive behaviors in *C. elegans* and mice is striking.

Thus, while the specific motor behaviors subject to repetitive engagement differ dramatically among humans, mice, and *C. elegans* (e.g. head banging, grooming, and reversals, respectively), our results suggest that a common underlying physiology may be at play. Our results also suggest that physiological and pathological repetitive behaviors may be appropriate settings in which to investigate the behavioral consequences of glutamate spillover in mammals.

Our study suggests intriguing similarities between the *C. elegans* CEPsh glia and mammalian astrocytes. Like vertebrate glia, the CEPsh cells are derived from neuroepithelial precursors, but are not neurons themselves[43]. The four CEPsh glia tile around the nerve ring, the brain neuropil of the animal, suggesting morphological similarities with vertebrate astrocytes tiling across the central nervous system[7,44]. Here, we demonstrate that CEPsh glia are structurally elaborate, projecting fine processes that penetrate the nerve ring and that can be found near synapses, as are vertebrate astrocyte processes. Our gene-expression profile also reveals gene-expression similarities between *C. elegans* CEPsh glia and vertebrate astrocytes. Furthermore, our studies exploring the function of the CEPsh glia-enriched GLT-1 transporter demonstrate similar roles to those of astrocyte-enriched GLT1/EAAT2 in preventing repetitive behavior[40]. Previously, we found that like mammalian astrocytes, CEPsh glia also regulate sleep behavior[15,45]. Thus, *C. elegans* is a suitable setting for investigating aspects of astrocyte function in neuronal circuit activities and behavior. Moreover, these findings emphasize the limitations of predicting behavior solely based on mapping of neuronal connectomes, and suggest that glial architecture around synapses and their molecular properties should be integrated into such models.

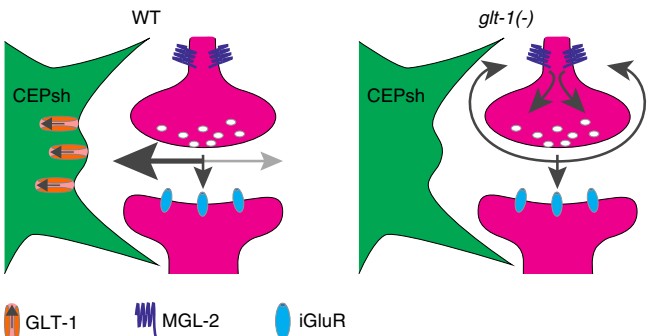

**Fig. 7** Model for feedback excitation by glutamate spillover inducing repetitive reversal behavior. Under normal conditions (WT) glutamate (indicated by the gray arrows) is rapidly cleared from the synaptic cleft by glia-expressed GLT-1. In *glt-1* mutants, glutamate is allowed to diffuse away from the synapse, resulting in activation of presynaptic MGL-2 and the induction of cycles of presynaptic activations

## Methods

**Strains**. *C. elegans* strains were cultured at 20 °C unless otherwise indicated. Wild-type (WT) animals were Bristol strain N2. Other strains used in this work are:
 OS1914: *nsIs105 (hlh-17*$_{pro}$::GFP)
 OS1965: *nsIs105; nsIs108 (ptr-10*$_{pro}$::myrRFP)

OS11314: *nsIs615 (glt-1*pro::GFP); *nsIs698 (mir-228*pro::nls-RFP, *unc-122*pro::mCherry)

OS10227: *nsEx5085 (hlh-17*pro::glt-1 cDNA::GFP, *elt-2*pro::mCherry); *nsEx3639 (glr-1*pro::nlg-1 cDNA::mCherry, *rol-6*); *glt-1(ok206)*

MT6308: *eat-4(ky5)*

OS3537: *nsIs168 (hlh-17*pro::recCaspase-3, *unc-122*pro::GFP); *nsIs108 (ptr-10*pro::myrRFP)[Line-1]

OS3549: *nsIs180 (hlh-17*pro::recCaspase-3, *unc-122*pro::GFP)[Line-2]

OS3540: *nsIs171 (hlh-17*pro::recCaspase-3, *unc-122*pro::GFP)[Line-3]

OS5111: *glt-1(ok206)*

OS9488: *glt-1(ok206); eat-4(ky5)*

OS5012: *nsIs168; eat-4(ky5)*

OS9470: *nsEx4876 (hlh-17*pro:: glt-1 cDNA:sl2:mCherry; *elt-2*pro::mCherry), *glt-1 (ok206)*

OS9464: *nsEx4872 (hlh-17*pro::glt-1 cDNA:sl2:mCherry; *elt-2*pro::mCherry), *glt-1 (ok206)*

OS10353: *nsEx5122 (myo*pro:: glt-1 cDNA:sl2:mCherry; *elt-2*pro::mCherry); *glt-1(ok206)*

OS10338: *nsEx5115 (myo*pro:: glt-1 cDNA:sl2:mCherry; *elt-2*pro::mCherry); *glt-1(ok206)*

OS9798: *nsEx5032 (eat-4*pro:: glt-1 cDNA:sl2:mCherry; *elt-2*pro::mCherry); *glt-1 (ok206)*

OS9799: *nsEx5033 (eat-4*pro:: glt-1 cDNA:sl2:mCherry; *elt-2*pro::mCherry); *glt-1 (ok206)*

OS10227: *kyEx4787 (rig-3*pro::iGluSnFR, *unc-122*pro::dsRed); *kyEx4786 (rig-3*pro::RCaMP1e, *unc-122*pro::GFP)

OS10567: *kyEx4786; kyEx4787; glt-1(ok206)*

OS11082: *mgl-2(tm355); kyEx4786; kyEx4787*

OS11091: *mgl-2(tm355); glt-1(ok206); kyEx4786; kyEx4787*

OS11090: *mgl-1(tm1811); kyEx4786; kyEx4787*

OS11086: *mgl-1(tm1811); glt-1(ok206); kyEx4786; kyEx4787*

OS11090: *mgl-3(ok1766); kyEx4786; kyEx4787*

OS11086: *mgl-3(ok1766); glt-1(ok206); kyEx4786; kyEx4787*

UL2838: *F45H11.4::GFP (mgl-2* transcriptional fusion)

OH13645: *otIS518 (eat-4*[fosmid]::sl2-mCherry:H2B, *pha-1*[ + ]); *him-5*

OS11749: *nsEx6012 (tdc-1*pro::mgl-2 cDNA:sl2:mCherry; *elt-2*pro::mCherry); *kyEx4787; kyEx4786; glt-1(ok206); mgl-2(tm355)*

OS12266: *nsEx6012 (tdc-1*pro::mgl-2 cDNA:sl2:mCherry; *elt-2*pro::mCherry)

OS11776: *nsEx6017 (tdc-1*pro:: mgl-2 cDNA:sl2:mCherry; *elt-2*pro::mCherry); *kyEx4787; kyEx4786; glt-1(ok206); mgl-2(tm355)*

OS12267: *nsEx6017 (tdc-1*pro:: mgl-2 cDNA:sl2:mCherry; *elt-2*pro::mCherry)

OS12275: *nsEx6169 (inx-1*pro:: mgl-2 cDNA:sl2:mCherry; *elt-2*pro::mCherry); *kyEx4787; kyEx4786; glt-1(ok206); mgl-2(tm355)*

OS12276: *nsEx6170 (inx-1*pro:: mgl-2 cDNA:sl2:mCherry; *elt-2*pro::mCherry); *kyEx4787; kyEx4786; glt-1(ok206); mgl-2(tm355)*

CX10295: *kyEx2418 (tdc-1*pro::ChR2-H134R::mCherry; *elt-2*pro::GFP)

OS11670: *kyEx2418; kyEx4786; glt-1(ok206)*

OS11671: *kyEx2418; kyEx4786;glt-1(ok206); mgl-2(tm355)*

OS11669: *kyEx4787; kyEx4786; egl-30(ad806)*

OS11777: *kyEx4787; kyEx4786; egl-30(ad806); glt-1(ok206)*

OS12268: *nsEx6164 (tdc-1*pro:: egl-30 cDNA:sl2:mCherry; *elt-2*pro::mCherry) *kyEx4787; kyEx4786; egl-30(ad806); glt-1(ok206)*

OS12282: *nsEx6164 (tdc-1*pro:: egl-30 cDNA:sl2:mCherry; *elt-2*pro::mCherry) *kyEx4787; kyEx4786; egl-30(ad806)*

OS12272: *nsEx6167 (tdc-1*pro:: egl-30 cDNA:sl2:mCherry; *elt-2*pro::mCherry) *kyEx4787; kyEx4786; egl-30(ad806); glt-1(ok206)*

OS12283: *nsEx6167 (tdc-1*pro:: egl-30 cDNA:sl2:mCherry; *elt-2*pro::mCherry) *kyEx4787; kyEx4786; egl-30(ad806)*

OS11668: *kyEx4787; kyEx4786; goa-1(n1134)*

OS11778: *kyEx4787; kyEx4786; goa-1(n1134); glt-1(ok206)*

OS11908: *nsEx6036 (tdc-1*pro:: mgl-2 cDNA:GFP; *unc-122*pro::GFP); *nsEx3639*

Germline transformations were performed using micro-injection techniques[46]. Stable transgenes were obtained via psoralen integration[47].

**DNA for gene expression**. For cell-specific gene expression, the following promoter regions were used. DNA fragments were PCR amplified from *C. elegans* genomic DNA or existing plasmids:

*hlh-17*- A 4 kbp region upstream of the ATG
*glr-1*- A 5.3 kbp region upstream of the ATG
*myo-3*- A 2.4 kbp region upstream of the ATG
*eat-4*- A 4 kbp region upstream of the ATG
*glt-1*- A 4 kbp region upstream of the ATG
*tdc-1*- A 1.1 kbp region upstream of the ATG

For GLT-1 expression, first strand cDNA of *glt-1* isoform1 was prepared from total *C. elegans* mRNA using SuperScriptIII (Invitrogen). cDNA was PCR amplified, ORF was sequenced after cloning to verify that no mutations were introduced. Similar approach was used to clone MGL-2 cDNA. Plasmid containing neuroligin fused to mCherry was a gift from Dr. Kang Shen.

**Postembryonic cell ablation**. For genetic ablation of the CEPsh glia, the two parts of a reconstituted Caspase-3[48] were expressed in CEPsh glia starting in early L1 larvae using the *hlh-17* promoter[49]. Ablation was determined by lack of *ptr-10*pro::myrRFP and *hlh-17*pro::GFP expression in the CEPsh glia (Supplementary Fig. 2a, b), and confirmed in three animals by EM reconstruction, which also demonstrates intact nerve-ring structure and organization (Supplementary Fig. 2c).

**Fluorescence and electron microscopy**. Continuous optical sections images at 0.2 μm spacing were collected and deconvolved on a Deltavision Core imaging system (Applied Precision) with a UPLSAPO ×60 Silicon oil objective (Olympus) and a pco.edge sCMOS camera (PCO AG) (Rockefeller University Bio-Imaging Resource Center). Maximum brightness projections of contiguous optical sections were obtained using ImageJ. In Supplementary Figs. 2b and 7a, red channels gamma was corrected (0.5) to allow observing features of both low and high intensities. 3D reconstruction was done using Imaris (Bitplane). Alternatively, imaging was performed using an Axioplan II fluorescence microscope (Zeiss) equipped with an AxioCam camera. Transmission electron microscopy was carried out on animals prepared and sectioned using standard methods[50]. Samples were imaged using an FEI Tecnai G2 Spirit BioTwin transmission electron microscope equipped with a Gatan 4K × 4K digital camera (Rockefeller University Electron Microscopy Resource Center). Figures were assembled using Photoshop (Adobe Software).

**Glutamate and calcium imaging from AVA**. Imaging of glutamate input to, and calcium signals in AVA neurons, in first-day adult animals, were performed in animals immobilized using 4 mM tetramisole in a microfluidic device[25]. Animals were provided with food cues continuously from the recording onset to lower basal neuronal activities; this was followed by two short cycles of food cue removal (20/10 s each) to verify that the recorded animals have normal induced responses. Food cues were obtained by growing OP50 *E. coli* bacteria in NGM-liquid buffer overnight at 37 °C. Bacteria were then removed by centrifugation at 4000 RPM for 5 min, and conditioned media collected and cleared through a 0.22 μm filter. Images captured at one frame per second using a Photometrics CoolSnap HQ2 camera (Roper Scientific), connected to a dual channel imaging system (DV-2; Photometrics), with 595 nm filter to separate green and red emission, and mounted on an Examiner A1 microscope with a ×40 1.3 NA Apochromat oil objective (Zeiss). AxioVision 4.7 (Zeiss) software was used for movie acquisition and ImageJ was used for signal intensity measurements from the AVA neuron (calcium) and its nerve-ring process (glutamate; Supplementary Movies 7, 8). The resulting fluorescence traces were background corrected and subsequently analyzed with custom-written MATLAB scripts (see below).

For detection of spontaneous activity in dual glutamate and calcium recordings from AVA, we first determined whether an animal was responsive to the food cue, using two food cue cycles. To this end, we removed shot noise from the fluorescence traces with a Gaussian low-pass filter (standard deviation, 1 s) and normalized the smoothed signals such that their minimum value was equal to zero and their maximum value was equal to one. Then, we computed the first derivative of the smoothed and normalized signals as,

$$\frac{\mathrm{d}F}{\mathrm{d}t}(t_n) = \frac{F(t_n) - F(t_{n-1})}{2\Delta t}. \tag{1}$$

A worm was considered responsive to the food cue if the derivative trace exceeded a value of 0.05 within ±5 s from food cue removal. Only worms that responded to either or both food cue removals were considered for further analysis of spontaneous activity.

The analysis of spontaneous activity was based on the calcium signal from AVA, because calcium signals were much stronger and exhibited higher signal-to-noise ratio. Note, however, that the AVA glutamate signal in responsive animals was usually highly correlated with the calcium signal throughout all genotypes (see main text).

The background subtracted spontaneous calcium activity traces and subsequently acquired calcium activity traces during food cue removals were first filtered with a Gaussian low-pass filter (standard deviation, 0.5 s) to remove shot noise and then the minimum and maximum signal intensity over the entire recording period was determined. Then the raw spontaneous calcium activity traces were normalized such that the minimum of the overall signal was zero and the maximum of the overall signal was 1. The min/max-normalized spontaneous activity traces were then filtered with a Gaussian low-pass filter (standard deviation, 1 s) and we computed the first derivative of the smoothed and min/max-normalized signals. At the beginning of the 540 s recording periods, most animals showed no or little spontaneous calcium activity. An animal was classified as initiating spontaneous activity if the first derivative signal at any point during the 540 s recording period prior to the two food cue cycles fulfilled one of the following two criteria: (i) the first derivative exceeded a threshold of 0.15 at any time during the recording, (ii) the first derivative exceeded a threshold of 0.05 for two or more consecutive frames. The time when the activity trace first fulfilled one of the two criteria was considered the time of spontaneous activity onset after applying the food cue.

Animals with spontaneous calcium activity showed highly dynamic time traces with transient oscillations of varying frequency and intensity in time but still

exhibited common features among animals of the same genotype. To characterize these features, we applied wavelet analysis to calcium signals, which more accurately captures localized temporal and frequency information than Fourier analysis. We used complex-valued Morlet wavelets composed of a Gaussian envelope and a plane wave

$$\phi(t, f) = \sqrt{\frac{\pi}{\sigma}} e^{-t^2/2\sigma^2} e^{i2\pi ft}. \tag{2}$$

The wavelet spectrum $W(t, f)$ of a signal $S(t)$ is then given by

$$W(t, f) = \int dt' \phi(t - t', f) S(t'). \tag{3}$$

For a given frequency $f$, the width of the Gaussian was chosen as

$$\sigma = \frac{\xi}{2\pi f}, \tag{4}$$

with $\xi = 4$. This parameter choice enables detection of local frequency variations on scales larger than two periods while at the same time enabling robust estimation of the local frequency itself. Normalization with $1/\sqrt{\sigma}$ ensures that wavelet coefficients can be compared across frequency ranges. We analyzed frequencies from 0.01 to 0.2 Hz. Edge effects of the wavelet transformations were minimized by padding the original signal with reflected signals on both ends.

To determine common temporal features of spontaneous activity traces for a given genotype, we first time-averaged the wavelet spectrum of each spontaneously active animal from the onset of spontaneous activity to the end of the 540 s recording period (see Fig. 4d, f). To obtain the mean time-averaged frequency amplitudes (e.g. Fig. 4g), we then averaged these time-averaged wavelet spectra over all animals of a given genotype, weighing each animal with the fraction of time it was spontaneously active during the recording period.

For heatmap plots of spontaneous glutamate and calcium activities, raw fluorescence traces were normalized such that their minimum value was equal to zero and their maximum value was equal to one:

$$F_{norm} = \frac{F(t) - F_{min}}{F_{max} - F_{min}}. \tag{5}$$

Plots were generated by an R script and include only the first 9 min of recordings, where the food cue was continuously present.

For correlation between animal head motion and AVA calcium signals, first-day adult animals were recorded for 8 min at 1 frame per second in the presence of food cue, without the addition of anesthesia in a microfluidic device. Head position was obtained through alignment of the position of the AVA neuron between consecutive frames using the ImageJ Template Matching plugin, and extracting the difference in position between consecutive frames on the $x$-axis (motion on the $y$-axis is limited for animals trapped in the microfluidic chamber). Both the head position signal and the calcium signal were first Gaussian low-pass filter (standard deviation, 3 s for position, 1.5 s for calcium). We then normalized the calcium signal to unit variance, calculated its first derivative and determined the times when the first derivative crossed a threshold of 0.1 from below or a value of $-0.1$ from above. These time points were considered transitions between high and low calcium states (magenta and green dots in Fig. 4i). We calculated the overall position change during each high and low calcium state. If overall position increased, the animal was considered moving forward during a given calcium state and vice versa. Correlation between forward/backward motion and up/down states was assessed with Pearson's phi coefficient.

**Analysis of worm locomotion**. Twenty to forty first-day adult animals were placed on an NGM plate (1 mM NaCl; 1 mM CaCl; 1 mM MgSO₄; 1.7% agar; 0.25% Peptone; 5 µg/ml Cholesterol; 25 mM KPO₄, pH 6.0) with no food, and habituated for 20 min. Then their motion was recorded for 30 min at two frames per second using a video camera (Basler ace acA3800; Supplementary Movies 1–3). Thereafter, animal locomotion was analyzed using custom software written in Java[15]. The shape of a worm's body is used to discriminate between reversals and other direction changes like omega turns. Body shapes are not fully resolved in our images; thus, to assess the shape of a worm's body we compute its inertia matrix. This amounts to fitting an ellipse to the body, and the elongation of the body, used in the following to identify turning worms, is defined as the long axis of the ellipse.

To explicitly allow for uncertainties in the estimation of a worm's position and the identification of discrete events in its movement, trajectories are analyzed using a Hidden Markov Model[51]. As described below, the model is comprised of a set of discrete states such as crawling and turning. At each time step, transitions between these states can occur with a certain probability. The sequence of states corresponding to a track is inferred from the observed movement of the worm, i.e. the position and shape of its body in successive frames. The inference is probabilistic, i.e. for every frame, each state is assigned a fractional probability. We note that the model is not intended as a fully accurate representation of worm behavior. Rather, it is meant to allow a well-defined quantification of events with allowance for uncertainty.

To define the orientation of worm movement (backward or forward), each trajectory is projected along a one-dimensional coordinate $x$. The increment $\Delta x$ of this coordinate between successive frames is obtained by projecting the

displacement of the worm along the axis of its body. The orientation of the body (head vs. tail) is not visible in the images; thus, we arbitrarily choose the direction along the body axis that corresponds to positive increments in $x$. This choice is made for the first frame, then we keep track of the orientation such that the relation between changes in $x$ and direction of worm movement remains consistent. Whether forward movement corresponds to increasing or decreasing $x$ is determined as part of the inference.

The model comprises four behavioral states: forward crawling, backward crawling, forward turn, and omega turn. For the purpose of tracking the orientation of the body, the forward and backward crawling states are each subdivided into two states, depending on whether increasing $x$ corresponds to forward or backward movement (labeled + and −). We do not attempt to keep track of the body orientation in the turning states, and they admit no such subdivision. In total, the model thus has six states (Supplementary Fig. 1a). Not all possible transitions are permitted: for instance, the sign (+ or −) must remain the same when the worm reverses, and an omega turn occurs only in the sequence backward crawling → omega turn → forward crawling.

For the inference of model states, we specify probability distributions for the observable variables in each state. Crawling and turning states are distinguished by the relative elongation $r$ of the body, defined as the ratio between the instantaneous elongation of the body and its average over a track. For a worm that is crawling forward or backward, $r$ is close to 1. When the worm makes a forward or omega turn, it bends its body and $r$ is less than one. Under the model, the distribution of $r$ for crawling worms is a Gaussian distribution with average one and standard deviation $\sigma$:

$$P(r|\text{crawling}) = \frac{1}{\sigma\sqrt{2\pi}} e^{-\frac{(r-1)^2}{2\sigma^2}}, \tag{6}$$

where $\sigma = 0.075$ is close to the variance of the observed distribution in moving worms.

When the worm is turning, we simply take the distribution to be uniform over the interval [0,1]. For numerical reasons it is desirable for the distribution to be non-zero for $r > 1$, and we take the distribution to decay as a Gaussian with the same standard deviation $\sigma$ as above:

$$P(r|\text{turning}) = \begin{cases} 1 \text{ if } r \leq 1 \\ e^{-\frac{(r-1)^2}{2\sigma^2}} \text{ if } r > 1 \end{cases}. \tag{7}$$

Forward and backward movements are distinguished according to the increment $\Delta x$ of a worm's position between frames. Equivalent to specifying distributions for $\Delta x$ in each state, we specify probabilities for each state given a value of $\Delta x$. Allowing for uncertainty in the position of the worm, we require the displacement to be large enough to score the direction of movement with confidence:

$$P(\text{forward}^+|\Delta x) = P(\text{backward}^-|\Delta x) = \sigma\left(\frac{\Delta x}{c\Delta t}\right) \tag{8}$$

and

$$P(\text{forward}^-|\Delta x) = P(\text{backward}^+|\Delta x) = 1 - \sigma\left(\frac{\Delta x}{c\Delta t}\right), \tag{9}$$

where $\sigma$ is the sigmoidal function

$$\sigma(x) = \frac{1 + \text{erf}(2x)}{2} \tag{10}$$

and the velocity $c = 0.1$ mm s⁻¹ is chosen to be a fraction of the typical velocity of moving worms (about 0.25 mm s⁻¹). Thus, a worm must travel a finite distance in both directions for a reversal to be identified with confidence, and small fluctuations in the estimated position of a motionless worm are discounted.

Given a matrix of transition probabilities, the forward-backward algorithm[51] allows efficient computation of the likelihood of state sequences and other quantities of interest, such as the number of transitions between pairs of states. The transition probabilities can thus be re-estimated from the output of the algorithm. The Baum-Welch algorithm[51,52] iterates rounds of the backward-forward algorithm and parameter re-estimation; the likelihood of the data under the model increases monotonically at each step, guaranteeing convergence to a maximum. Thus, the transition probabilities are not free parameters of the model but estimated from the data.

To infer the absolute direction of movement, we use the fact that periods of backward movement are brief, much shorter on average than periods of forward movement. If we simply initialize the transition probabilities with a higher rate for transitions from backward to forward than from forward to backward, the above inference procedure converges to a set of transition probabilities and state probabilities that assigns the direction (increasing or decreasing $x$) corresponding to longer bouts of crawling to forward movement with high probability (Supplementary Fig. 1b−d).

To characterize the duration of forward bouts, periods of backward or forward movement are defined as periods when the inferred probability of the backward crawling state is greater or smaller than one half, respectively. The distribution of forward bout durations cannot be measured directly, because some tracks are lost when the worm exits the field of view or touches another worm. The distribution is thus estimated from the rate of change between forward and backward movement, as

a function of time since the beginning of the bout. If $p_{f \to b}(i)$ denotes the probability that a worm that moved forward for $i$ time steps moves backward in the next step, the probability $p_f(n)$ that a forward bout lasts at least $n$ steps is estimated as

$$p_f(n) = \prod_{i=0}^{n-1}(1 - p_{f \to b}(i)). \qquad (11)$$

Repetitive reversals were quantified based on the deviation between the observed distribution of forward bouts and the exponential distribution associated with a Poisson process. To do so, the reversal rate $r(t)$ observed after a time interval $t$ moving forward was modeled as

$$r(t) = r_0 + \delta r e^{-\frac{t}{\tau}}, \qquad (12)$$

where $r_0$ denotes a baseline, long-time reversal rate, and $\delta r$ is an enhancement in the reversal rate at short times, decaying over a time $\tau$. The corresponding probability $p_f(t)$ that a worm is still moving forward after a time interval $t$ is given by

$$\log p_f(t) = -r_0 t - \delta l \left(1 - e^{-\frac{t}{\tau}}\right), \qquad (13)$$

where $\delta l = \tau \delta r$. This minimal model for the transient enhancement of reversals following an initial reversal fits well with the behavior observed in $glt$-$1$ mutants, with a time scale $\tau \approx 20$ s, and provides a natural definition of a repetitive behavior index

$$RI = \delta l / \log 2. \qquad (14)$$

An RI of 1 means that at long times, 50% fewer worms are still moving forward than would be expected according to the baseline reversal rate. Noting that a similar time scale $\tau$ is found in wild type (although the deviation from a constant reversal rate is smaller), we imposed a fixed value $\tau = 20$ s for all lines, to keep a consistent definition of early vs. late reversal rates, and fit $r_0$ and $\delta l$ independently for each experiment.

**Reversal response to mechanical stimuli.** To determine the response to mechanical touch[53], first-day adult animals were placed individually on 6 cm NGM-agar plates, freshly covered with a thin bacterial layer. Touch was applied to the anterior part of animals that are either in forward motion or paused states, using a thinly pulled glass capillary with a blunted end. Each animal was probed eight times, with at least 15-min interval between consecutive stimuli. Animals were scored blindly based to three criteria: no response, or short pausing (pause); reversal response composed of backward bout of about 2–3 body bends (Supplementary Movie 5); animals responding with multiple reversal events that are separated by a short pause or forward bout, or with a long reversal bout composed of multiple body bends (repetitive; Supplementary Movie 6). Response was completed once animals resumed forward motion for more than three body bends, or if animals stayed paused for more than 5 s.

**Reversal response to optogenetic stimuli.** L4 larvae were grown overnight on plates with OP50 *E. coli* bacteria with or without all-trans retinal (50 μM). The following day, 10–20 young adults were prepared for locomotion recordings as above. Animal locomotion was recorded for 10 min (two frames per second), and after a minute plates were illuminated with blue light ($\lambda = 455$ nm, intensity ~35 μW mm$^{-2}$, Mightex) for 5 s, followed by a 115 s recovery interval. This was repeated for a total of four stimulations. Reversal events were analyzed using the worm-tracking code, as above. Animals exposed to blue light may show no response, a single reversal response or multiple reversal events (repetitive reversals), during the short stimuli. To determine the fraction of repetitively responding animals, the total fraction of animals responding to the stimuli ($R_{total}$) and the fraction of animals with multiple responses ($R_{multiple}$) were calculated from movies of animals treated (+) or not-treated (−) with retinal. To define response specificity to light, the fraction of responses seen in retinal (−) animals was subtracted from the fraction of retinal (+) responsive animals, and the normalized fraction of animals reversing multiple times was calculated as

$$\text{Normalized } R_{multiple} = \frac{\left(R_{multiple+}\right) - \left(R_{multiple-}\right)}{\left(R_{total+}\right) - \left(R_{total-}\right)}. \qquad (15)$$

**FACS-based CEPsh cell isolation.** For isolation of the CEPch glia, animals expressing GFP in the CEPsh cells (OS1914) were grown on 48× NEP 10 cm plates seeded with NA22 *E. coli* bacteria. When plates are full with gravid animals, embryos are obtained by bleaching, followed by three washes with sterile M9 buffer in a tissue culture hood, and incubated rotating in $16 \times 25$ cm flasks containing CeHR medium[54] for 36 h at 20 °C. This results in a mixed population of stage 2–3 larvae (L2-3; about $3 \times 10^6$ animals). Animals were dissociated using SDS-DTT (0.25% SDS; 200 mM DTT; 20 mM HEPES, pH 8.0; 3% sucrose) and pronase E (15 mg/ml). We used 2:1 ratio of SDS-DTT to a volume of packed worms pellet, followed by 4 min incubation on ice. After washes, 4:1 ratio of Pronase E was added to the packed worms pellet and animals were incubated rotating at 20 °C for 5 min, followed by 10–15 min of gentle homogenization (7 ml dounce homogenizer, pestle clearance 0.02–0.056 mm, Kimble Chase). After washes with ice-cold egg buffer (1.18 M NaCl; 480 mM KCl; 20 mM CaCl$_2$; 20 mM MgCl$_2$; 250 mM HEPES, pH 7.3) to remove pronase E, cells were filtered through a

5 μM filter to remove undigested animal fragments, and immediately sorted by FACS.

CEPsh cell sorting was done using a BD FACS Aria sorter equipped with 488 nm laser (Rockefeller University Flow Cytometry Resource Center), with egg buffer as the sheath buffer to preserve cell viability. Dead cell exclusion was carried out using DAPI. Gates for size and granularity were adjusted to exclude cell aggregates and debris. Gates for fluorescence were established using wild-type (N2) nonfluorescent animals. 160,000–1,000,000 GFP-positive events were sorted per replicate, which represented 0.1−0.4% of total events (after scatter exclusion), which is roughly the expected labeled-cell frequency in the animal (~0.4%). GFP-negative events from the same gates of size and granularity, representing all other cell types, were also sorted for comparison. Cells were sorted directly into TRIzol LS (Ambion) at a ratio 3:1 (TRIzol to cell volume).

**RNA extraction and sequencing.** RNA was extracted from the cells following the TRIzol LS protocol guidelines, until the isoproponal precipitation step, then RNA was re-suspended in extraction buffer of a RNA isolation kit (PicoPure, Arcturus), and isolation continued according to the manufacturer's guideline. This two-step purification protocol helps obtaining RNA of high quality when starting with samples of large volumes, and resulted in a yield of around 8−24 ng per replica with RIN ≥ 8, as measured by a Bioanalyzer (Agilent). Specificity of the RNA was determined by using ~7% of purified RNA for first strand cDNA synthesis with SuperScript III kit (Invitrogen), followed by qPCR (LightCycler 480, Roche) analysis, using *lmn-1* as a control to standardized the samples, and *hlh-17* to measure differential expression between samples derived from the CEPsh glia to the control. All subsequent steps were performed by the Rockefeller University Genomics Resource Center. Briefly, mRNA amplification and cDNA preparation were performed using the SMARTer mRNA amplification kit (Clontech). Labeled samples were sequenced using an Illumina HiSeq 2000 sequencer using either 50- or 100-base read protocols.

**RNA-Seq quality and differential expression (DE) analyses.** Fastq files were generated with CASAVA v1.8.2 (illumina), and examined using the FASTQC (http://www.bioinformatics.babraham.ac.uk/projects/fastqc/) application for sequence quality. Reads were aligned to customized build genome that combine *C. elegans* WS244 reference genome (ftp://ftp.wormbase.org/pub/wormbase/releases// WS244/species/c_elegans/PRJNA13758/c_elegans.PRJNA13758.WS244. genomic.fa.gz) and GFP using the STAR v2.3 aligner with parameters[55] (--out-FilterMultimapNmax 10 --outFilterMultimapScoreRange 1). Mapping rate was >99% with >49 million uniquely mapped reads. The alignment results were evaluated through RNA-SeQC v1.17 to make sure all samples had a consistent alignment rate and no obvious 5′ or 3′ bias[56]. Aligned reads were summarized through featureCounts[57] with gene models from Ensemble (Caenorhabditis_elegans. WBcel235.77.gtf) at gene level unstranded: specifically, the uniquely mapped reads (NH "tag" in bam file) that overlapped with an exon (feature) by at least 1 bp on either strand were counted and then the counts of all exons annotated to an Ensemble gene (meta features) were summed into a single number. rRNA genes, mitochondrial genes and genes with length <40 bp were excluded from downstream analysis.

Experiment was done with three independent replicates. DESeq2 was applied to normalize count matrix and to perform differential gene-expression analysis, comparing RNA counts derived from the CEPsh cells (GFP positive) to RNA counts that were derived from all other *C. elegans* cells, using negative binomial distribution[58].

**Gene-expression comparison between *C. elegans* and mouse cells.** For gene-expression comparisons between *C. elegans* and mouse, RNA-sequencing data for the various mouse brain cells were downloaded from GEO (GSE52564), and fold-enrichment (FE) values were calculated as previously described[16]. RNA-sequencing data of oligodendrocyte precursor cells (OPC), newly formed oligodendrocytes (GC) and myelinating oligodendrocytes (MOG) were merged and averaged to generate a list of oligodendrocyte-expressing genes. A data set assigning orthologs between *C. elegans* and mouse genes was downloaded from PANTHER (ftp://ftp. pantherdb.org/ortholog/14.0/), including both least diverged orthologs (LDO, 5117 records) and all other orthologs (O, 16,487 records). To find unique identifiers for mouse genes, gene symbols from the RNA-sequencing data were converted into Mouse Genome Informatics (MGI) ID numbers to match the data from PANTHER. WormBase gene ID number was used as a unique identifier for *C. elegans* genes.

To perform heatmap hierarchical clustering of mouse brain cells and CEPsh glia-expressed genes, mouse genes were converted into their corresponding *C. elegans* orthologs based on the PANTHER database. Next, FE values in each list were converted to rank order. Gene ranking within each column was used as the input matrix and the heatmap was generated through R (https://cran.r-project.org/ web/packages/pheatmap) using correlation as the distance function.

To test for cell type-specific enrichment of the various mouse glia gene lists in the CEPsh glia-gene list, we used step-wise fold-enrichment (FE) cutoff values (for CEPsh-glia, log$_2$ FE = 1, 1.5, 2, 2.5; for mouse glia, log$_2$ FE = 1, 1.5, 2, 2.5, 3, 3.5, 4). To achieve gene-expression specificity for the various mouse glia, we only

considered genes that are uniquely present in a cell type for any given fold change cutoff value. Each *C. elegans* gene was counted as related to a mouse glial cell type if the *C. elegans* gene had any ortholog present on the list of enriched genes from the mouse brain RNA-seq data. For each fold change cut-off pair, we calculated the enrichment score[59]. Specifically, we compared the frequency of CEPsh glia genes with mouse brain-enriched orthologs to the frequency of *C. elegans* genes in the genome with mouse brain-enriched orthologs, normalized by the number of genes in each list with a mouse ortholog:

$$\text{Enrichment score} = \frac{\text{Proportion of CEPsh glia enriched genes with a mouse ortholog}}{\text{Proportion of worm genes with a mouse ortholog}}, \quad (16)$$

$$\begin{aligned}&\text{Proportion of CEPsh glia} - \text{enriched genes with a mouse ortholog}\\ &= \frac{\text{\# of CEPsh glia enriched genes with a mouse brain enriched ortholog}}{\text{\# of CEPsh enriched genes with a mouse ortholog}},\end{aligned} \quad (17)$$

$$\begin{aligned}&\text{Proportion of worm genes with a mouse ortholog}\\ &= \frac{\text{\# of worm genes with a mouse brain enriched ortholog}}{\text{\# of worm genes with a mouse ortholog}}.\end{aligned} \quad (18)$$

A $\chi^2$ test between the various mouse brain cells at the various FE cutoff values was used to correct a potential bias ortholog-mapping may have introduced, and the ratio between the expected value of the enrichment score (given there is no cell type specificity) and the observed enrichment score was calculated as the association score (see Supplementary Data 1).

**Statistics**. Unpaired two-sided Students' *t* test was used to determine the statistical significance between two groups, or between several samples compared to a control group with Bonferroni correction for multiple comparisons. For multiple comparisons ANOVA Tukey's HSD post hoc test was used, and in cases where the variance between samples were too high (Brown Forsythe test, $p < 0.01$), a Kruskal−Wallis one-way ANOVA with Dunn's multiple comparison test was used. Tests were done using Prism (GraphPad Software). For overrepresentation analysis Fisher's exact test was used. $\chi^2$ test was used to determine the fit of the observed cumulative probability to continue moving forward with an exponential model using a MATLAB script. The null hypothesis is that the value of residuals from the data points to fit an exponential curve is expected from the experimental noise. The weighted sum of squared residuals,

$$S_{\min} = \sum_{i=1}^{n} \frac{\Delta r_i^2}{\sigma_i^2} \quad (19)$$

the number of degrees of freedom ($n$) is defined by the number of binned time points, and the variance ($\sigma$) is defined by the standard deviation[60].

**Reporting summary**. Further information on experimental design is available in the Nature Research Reporting Summary linked to this article.

## Data availability
The RNA-seq data sets generated during the current study have been deposited in the European Nucleotide Archive (ENA) with the primary accession code PRJEB31134. The source data underlying Figs. 2a, c, 3a−f, 4i, 5a−c, 6a, b, and Supplementary Figs. 1e, 4a, b, 5a, 6a, b, 7b, 8a−c are provided as a Source Data file. A reporting summary provided for this article is available as a Supplementary Information file.

## Code availability
All customized codes described in this manuscript are available from the corresponding author upon request.

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

## Acknowledgements

We thank Daniel Colón-Ramos, Kang Shen, Cori Bargmann, Oliver Hobert, Alipasha Vaziri, Vincent O'Connor, Ian Hope and the *Caenorhabditis* Genetic Center (CGC) (NIH P40 OD010440) for strains and reagents; The Rockefeller University Bio-imaging, Flow Cytometry, Electron Microscopy, and Genomics Resource Centers for technical support; Sagi Levy and Alina Rashid for help with data analysis; Andrew Gordus, Nikolaos Stefanakis, Ido Amit, and Ivo Spiegel for experimental advice; and Shaham lab members for comments and discussion. This work was supported by an EMBO fellowship ALTF 870-2007 to M.K., a postdoctoral fellowship (LT000250/2013-C) from the Human Frontier Science Program (HFSP) to W.K., NSF grant PHY 1502151 to Eric Siggia, and by NIH grants R01NS073121, R01NS064273, R01NS095795, and R35NS105094 to S.S.

## Author contributions

M.K. and S.S. designed experiments, interpreted data and wrote the manuscript. F.C. developed the worm-tracking and locomotion analysis code. W.K. analyzed the glutamate and calcium kinetics studies. A.B. recorded neural glutamate and calcium signals. Y. Lu performed electron microscopy imaging. Y. Liang and A.S. conducted mRNA-sequencing analysis. M.K. conducted all other experiments.

## Additional information

**Competing interests:** The authors declare no competing interests.

