## [Peer Review File · Nature Communications]

Reviewers' Comments:

Reviewer #1:

Remarks to the Author:

In the current manuscript of Katz et al., the authors investigate the authors show a role of the glutamate transporter *glt-1* in neurotransmitter clearance in the nerve ring acting in CEPsh glia which show a number of similarities to mammalian astrocytes.

Using behavioral assays as well as in vivo glutamate and calcium imaging they show that loss of *glt-1* leads to spillover of glutamate after release from RIM neurons which activated extrasynaptic mGluR receptors on RIM itself, leading to repetitive glutamate release from RIM that activate AVA neurons which in turn control reversal behavior of the animals.

The manuscript is well written and the findings are novel and of interest to the neuroscience community. It further supports the similarities of *C. elegans* CEPsh glia to mammalian astrocytes and extends our knowledge of neuron glia interaction in the nervous system.

Comments:

Figure 1b and 1c about the hierarchical clustering of gene expression are not very clear to me. I don't know if there is more information in the supplement but from this clustering analysis I can't really tell which genes might be co-enriched in mouse astros and CEPsh glia. The only example they talk about is *glt-1*. Maybe presenting a couple more examples would illustrate their claim better.

They use FE (fold enrichment) as a measure but I can't tell if it is enrichment over a certain cell type or total gene expression?

They use *vglut* mutants (*eat-4*) to show that the *glt-1*/ CEPsh ablation phenotype is dependent on glutamate release, however *vglut* mutants on their own have dramatically reduced reversal rates compared to wt, calling the conclusions from this experiment somewhat into question (Fig. 2a).

That neuronal rescue of *glt-1* mutants is more efficient than CEPsh rescue is surprising and the authors should discuss why this might be and why they are still concluding that *glt-1* is acting CEPsh glia (Fig. 2C).

In Fig. 4 it is a bit surprising that absolute glutamate levels don't seem to be elevated in *glt-1* mutants. This is somewhat surprising since in the mouse brain loss of transport function leads to severe increases in extracellular glutamate. Maybe the authors could discuss this point.

Is it possible to show spillover of glutamate directly? For instance by expressing iGluSnfR in CEPsh glia or in RIM itself, activate RIM and compare glutamate signal spread in wt controls vs *glt-1* mutants?

The rescue data using *tdc-1pro::mgl-2* should include a control of this construct alone in a wt background to see what its effect on reversals is on its own (Fig. 5b).

While Fig. 5 a-c show that the mGluR mutant *mgl-2* (*tm355*) is able suppress the effect of *glt-1*, this effect seems to be partial. Either the mGluR mutant is not a null, has some perdurance (I am not sure about the crosses here) or alternatively the effect of *glt-1* on repetitive reversal behavior is not solely dependent on mGluR but might also act via some other mechanism. This should be discussed by the authors.

Reviewer #2:

Remarks to the Author:

In this manuscript, Katz et al report of a simple feedback process that can explain repetitive behavior in *C. elegans* worms. The underlying mechanism involves inactivation of the glutamate transporter *GLT-1* in CEPsh cells which results in glutamate spillover. The un-cleared glutamate, secreted by the RIM neuron, presumably rebinds the glutamate receptor *MGL-2*, expressed on RIM, to initiate another activation cycle. Postsynaptic to RIM is the command neuron AVA that is repeatedly activated

to promote a repetitive reversal behavior.

The authors performed a comprehensive rigorous work and meticulously analyzed the data applying the appropriate statistical tests. I am less certain about the novelty of the drawn conclusions as impaired glutamate signaling has already been established as a possible cause of various behavioral disorders, including repetitive behaviors in humans, such as OCD (see refs 34-35). Nevertheless, I find the simple autocrine mechanism, revealed in *C. elegans* worms, an intriguing and elegant demonstration of how aberrant glutamate clearance can lead to repetitive behaviors.

Specific comments for the authors:

Major:

1. The behavioral assays that score reversal rates (e.g. figs 2-3,5) are performed on plates with no food. Specifically, scoring starts 20 min. after the worms were placed on the plate. This is also the typical time in which worms switch from local food search (characterized by many reversals) to dispersal (long runs and less reversals). It is possible that defective glutamate signaling affects worms' sense of starvation, and consequently, the timing of this behavioral switch which might affect reversal rates when compared to WT animals. To exclude the possibility that starvation processes bias the reversal rates, the authors ought to repeat these experiments on plates with food. At least to verify that *glt-1* mutants still show enhanced reversal rates compared to WT and that GLT-1 rescue in CEPsh glia is similar to WT. In such experiments, scoring should start 2-3 after the transfer of the animals to the assay plate. Such conditions will also match the conditions used for functional analyses of Ca and Glu release (fig 4).
2. In line 184, the authors state that: "Wild-type animals initially show no glutamate release or calcium induction events in our assay. However, after 4 min \pm 24 sec of habituation to the food cue, on average, spontaneous glutamate release events onto AVA ensue". Looking at suppl. fig 4, it shows that this is not correct and at least half of the animals do show activity in the first 4.5 minutes. Actually, based on the suppl. figure, contrasting WT (fig 4a) and *glt-1* (fig 4b) is misleading as these are not the typical representative traces.
3. In fig. 5a, the repetitive behavior index of *glt-1(ok206)* is nearly 1.2 while in fig 3d the value is 0.7. I guess these values come from two independent experimental sets, but the difference seems a bit high. For example, if the value was 0.7 in fig 5a then it would not be significantly different from the double mutant *glt-1;mgl-2* as shown in fig 5a. It is particularly important to verify this issue since it is shown that the reversal rates of *glt-1* and *glt-1;mgl-2* are similar (suppl. fig 6a).
4. On a related note, in fig 5a RI of *glt-1* is significantly higher than *glt-1;mgl-2*. However, in fig 5b, if I read correctly the notations above the bars, then the RIs of these two strains are not significantly different. How come?
5. When RIM was light activated, the authors scored quick reversals during the short light stimulus (5 seconds as noted in methods line 683). This is very different from the oscillatory responses and repetitive behavior described in all other experiments, which were typically \sim 25 seconds. What is the reason that the scoring was done differently? According to the suggested model, I would expect to find the 25s period of oscillations following a single light activation of RIM. The authors should assay that in order to support their claims and the suggested model where the feedback through RIM underlies the observed repetitive phenotypes.
6. The authors show that RI is negligible in *egl-30/Gaq;glt-1* mutants but significantly increases in *goa-1/Gao;glt-1* mutants (as well as glutamate secretion and AVA activation events). However, these genes are widely expressed in the nervous system and many neurons could collectively contribute to the observed phenotypes. Thus, unless the genes are introduced to rescue in a neuron-specific manner it is practically impossible to tell whether RI and AVA activations are directly affected by MGL-2-mediated Ga signaling in the RIM neuron.
7. Ca and Glu traces are oscillatory and highly correlative. However, if the proposed model is that

Glu is not cleared from synaptic and extra-synaptic sites, should not glutamate levels remain high, or possibly, only mildly decrease? It seems that glutamate concentrations always return to basal levels (fig 4b).

8. Related to the above, could the authors explain why spillover-based oscillatory periods are ~25 seconds long? Naively, I would expect that following RIM activation, glutamate will remain in excess at synapses and extra-synaptic regions and instantly induce MGL-2 mediated RIM reactivation (similar to the quick reversals observed following light-activation of RIM).

Minor :

- Supplementary Fig. 1e: It is not clear what each of the three CEPsh-glia-abl bars denotes. Also, why reversal rate values of WT and ablated animals are markedly different between suppl fig 1e and fig 2a (~two-fold ratio)?

- Supplementary Fig. 3 legend: typo in 'intersept'.

Reviewer #3:

Remarks to the Author:

The authors use *C. elegans* to probe the role of glutamate signaling, particularly glutamate spillover at the RIM::AVA synapse in affecting animal behavior. They identify a conserved glial glutamate transporter, *glt-1* that clears glutamate at this synapse. Specifically, they show that *GLT-1* mutants fail to clear this synapse, causing activation of *mgl-2* on the RIM axon. This forms a positive feedback loop, leading to repetitive reversals. I do have a couple of concerns that would improve their manuscript (listed in no particular order).

Major concerns

1. It is possible that *MGL-2* might also act in RIM dendrites. In this case, the interesting synapse is between RIM::neuron (not AVA). Further, the altered RIM activity then modifies AVA activity through traditional neurotransmission.
2. Is glial *glt-1* really important? It looks like neuronal *glt-1* might also have a role in affecting neural activity and behavior (Fig. 2c). Perhaps tissue selective knockdowns could be used to sort out the role of the two transporters.
3. Statistics. It looks like the authors are assuming normality for all statistical measures. It is probably fine for most of the data, except for reorientations (a naturally bounded metric). This is particularly problematic when comparing genotypes with small numbers of reorientations.
4. The authors claim that glutamatergic synapses are close to glial *GLT-1*, but this is not clear (Figure 2b). Perhaps, they could add an additional synapse to their analysis.
5. It looks like the wavelet analysis shown in Fig. 4cdef wherein *glt-1* mutants have higher frequency oscillations at various time points seem to be derived from 1 animal? If so, I would recommend adding a few more to the analysis.
6. The authors argue that *MGL-2* function in RIM. I would recommend knocking down *MGL-2* in RIM specifically to test their hypothesis. It is possible that *MGL-2* acts in additional neurons to affect AVA function.

Minor Concerns

1. Figure 1b. Y-axis is confusing. Maybe there is a better way to present this data.
2. Figure 3c is a bit confusing. I thought that the plot shown here was the observed-exponential fit?
3. Figure 4i feels out of order. I would suggest that animal movement is correlated with calcium

transients.

4. Statistics in Figure 5c.

Response to reviewers

We thank the reviewers for their important and insightful comments on our manuscript. We have now performed several new experiments to address reviewer concerns, and have added additional discussion as suggested by the reviewers. We believe that the revised manuscript is now substantially strengthened, providing a comprehensive view of the synaptic mechanisms by which glutamate spillover induces repetitive behavior.

Bellow we address in detail all reviewer comments:

Reviewer #1 (Remarks to the Author):

In the current manuscript of Katz et al., the authors investigate the authors show a role of the glutamate transporter *glt-1* in neurotransmitter clearance in the nerve ring acting in CEPsh glia which show a number of similarities to mammalian astrocytes.

Using behavioral assays as well as in vivo glutamate and calcium imaging they show that loss of *glt-1* leads to spillover of glutamate after release from RIM neurons which activated extrasynaptic mGluR receptors on RIM itself, leading to repetitive glutamate release from RIM that activate AVA neurons which in turn control reversal behavior of the animals.

The manuscript is well written and the findings are novel and of interest to the neuroscience community. It further supports the similarities of *C. elegans* CEPsh glia to mammalian astrocytes and extends our knowledge of neuron glia interaction in the nervous system.

Comments:

1. Figure 1b and 1c about the hierarchical clustering of gene expression are not very clear to me. I don't know if there is more information in the supplement but from this clustering analysis I can't really tell which genes might be co-enriched In mouse astros and CEPsh glia. The only example they talk about is *glt-1*. Maybe presenting a couple more examples would illustrate their claim better.

The genes that were used for the analysis in Fig. 1c, including their *C. elegans* homologs and fold-enrichment values in different cell-types, are listed in Supplementary Table 1b.

To avoid confusion, we now also include in the revised manuscript a table with representative genes highly enriched in both *C. elegans* CEPsh glia and mouse astrocytes (Fig. 1d).

2. They use FE (fold enrichment) as a measure but I can't tell if it is enrichment over a certain cell type or total gene expression?

The FE value of each gene is the ratio between the mean number of mRNA reads obtained from CEPsh glia and the mean number of mRNA reads in all other cells (excluding the CEPsh).

To clarify this, the FE value is now explained both in the results and methods sections of the revised manuscript.

3. They use vglut mutants (*eat-4*) to show that the *glt-1*/ CEPsh ablation phenotype is dependent on glutamate release, however vglut mutants on their own have dramatically reduced reversal rates compared to wt, calling the conclusions from this experiment somewhat into question (Fig. 2a).

Our reasoning in these experiments is that if *eat-4*; *glt-1* double mutants still exhibited aberrant reversal rate, then *glt-1* effects cannot be due to extracellular glutamate. Since this is not the case, it is therefore possible that *glt-1*(-) mutant defects arise from excess extracellular glutamate. Nonetheless, another model, implied by the reviewer, is possible, in which *glt-1* and *eat-4* function in parallel in unrelated pathways, and in which disruption of either pathway is sufficient to block reversal behavior. For example, perhaps there are two neurotransmitters for reversal behavior, glutamate, and another one that depends on *glt-1*, and knocking out even one of these transmitters blocks reversals. While this is a formal possibility, we believe that this is highly

unlikely. GLT-1 protein has been previously shown to regulate glutamate transport ¹. Furthermore, the defects in *glt-1* mutants can be alleviated by mutating MGL-2, a glutamate receptor (Figs. 4g,h and 5a,b,c). From our glutamate imaging studies, the etiology of repetitive behavior in *glt-1* mutants is likely aberrant cyclic glutamate release from presynaptic neurons. Furthermore, when we record glutamate levels from *eat-4; glt-1* double mutants, we do not see glutamate release (data not shown), and also see no aberrant repetitive behavior. Taken together, therefore, we believe that the *eat-4; glt-1* double mutant provides corroboration that the effects of *glt-1* are related to synaptic glutamate release.

We have now clarified this point in the revised manuscript.

4. That neuronal rescue of *glt-1* mutants is more efficient than CEPsh rescue is surprising and the authors should discuss why this might be and why they are still concluding that *glt-1* is acting CEPsh glia (Fig. 2C).

This is an important point that we would like to clarify. As we discuss in our original manuscript, analysis of expression of a *glt-1* promoter::GFP transgene reveals only strong expression in CEPsh glia and weaker expression in head muscles. No expression is detected in neurons. This is corroborated by our RNA sequencing data, which shows a large enrichment of GLT-1 expression in glia over non-glial cells (many of which are neurons and muscles).

Regardless of the expression site, the main point we are trying to make is that *glt-1* affects extracellular levels/dynamics of glutamate released from synapses. To test this hypothesis, we sought an artificial way to express GLT-1 at glutamatergic synapses by ectopic expression in neurons. This indeed shows the most efficient rescue, suggesting that synaptic proximity likely plays a role in the effects of GLT-1 (Fig. 2c).

We have now modified the text to make these ideas more clear.

5. In Fig. 4 it is a bit surprising that absolute glutamate levels don't seem to be elevated in *glt-1* mutants. This is somewhat surprising since in the mouse brain loss of transport function leads to severe increases in extracellular glutamate. Maybe the authors could discuss this point.

This is also an important point we would like to clarify. The glutamate sensor (iGluSnFR) we use to detect extracellular glutamate is variably expressed between individuals, and therefore obtaining absolute measurements of glutamate levels is not possible. Because we are interested in glutamate dynamics, and not absolute levels, we normalize each activity trace according to the minimum and maximum fluorescent signal intensities observed in a recording interval, which simplifies computational detection of oscillations. The minimum value is defined as 0 and the maximum value is defined as 1. Therefore, the data presented in the manuscript cannot be used to deduce absolute glutamate levels.

Regarding the mouse studies, while microdialysis was previously used to measure overall extracellular glutamate concentration in animals blocked for glutamate transport ², synaptic glutamate levels and changes in these levels (i.e. oscillations) were not studied. Thus, the *C. elegans* and mouse data are not contradictory.

Indeed, we believe our original data (Fig. 2a) in fact suggests an overall increase in extracellular glutamate in *glt-1* mutants. Specifically, we compared reversal rates of wild type, *glt-1* mutants, and CEPsh glia-ablated animals, all also carrying a mutation in *eat-4/vGluT* (drastically reducing, but not completely eliminating glutamate secretion as there are other EAT-4-like proteins in *C. elegans*, and glutamate can also be released by unregulated secretion from cells). We found that in this sensitized background, both CEPsh glia ablated animals and *glt-1* mutants have increased (but still very low, see y axis in graph below and Fig. 2a) reversal rates (note that we are scoring average reversal rates and not repetitive reversals, as these are not seen in these strains). This suggests an increased level of extracellular glutamate. Furthermore, we find that *glt-1* mutants

have increased reversals compared to glia-ablated animals. An explanation for this may be that while extracellular glutamate is increased in both *glt-1* and CEPsh glia ablation animals, in CEPsh glia ablated animals, the physical barrier provided by glia is gone, allowing glutamate to diffuse more rapidly. Diffusion is attenuated in *glt-1* mutants, with an intact barrier, explaining the increased reversal rate.

Although we find these observations intriguing, we believe that discussing these tangential results would reduce the clarity of what is already a very busy manuscript. We therefore opt not to discuss this data in the paper.

6. Is it possible to show spillover of glutamate directly? For instance by expressing iGluSnfR in CEPsh glia or in RIM itself, activate RIM and compare glutamate signal spread in wt controls vs *glt-1* mutants?

This is an excellent proposal, which we would be very excited to test, and which we have been thinking about deeply. Unfortunately, to do the experiment properly is very challenging for two main reasons:

1) For such measurements to be quantitatively meaningful requires knowledge of the precise size and architecture of the extracellular space near release sites. This is currently beyond the capabilities of any microscope system, as what is needed is a merging of fast dynamic imaging (at frame rates of >1,000 Hz, as these are the rates of glutamate diffusion in brain slices based on modeling and electrophysiological data ³) in three dimensions with sub-diffraction-limited resolution. This is why we believe such studies have not been pursued in the mouse, where glutamate spillover has been studied for decades.

2) Perhaps even more important, the RIML and RIMR neurons have 54 and 50 output synapses, respectively (<http://wormwiring.org>). Some of these are physically very close, as observed by EM. Therefore, optogenetic activation of one of these neurons would likely generate glutamate release at multiple adjacent sites, making it difficult to determine if the signal being measured indicates spreading from a single site or mixing of discrete release events.

7. The rescue data using *tdc-1pro::mgl-2* should include a control of this construct alone in a wt background to see what its effect on reversals is on its own (Fig. 5b).

This is a great suggestion, and we performed the requested study. We found that MGL-2 overexpression in wild-type animals results in a significant increase in repetitive reversals. This is a very exciting result, as it further supports our model that MGL-2 is an intimate component of the machinery that generates repetitive behavior, and that MGL-2 levels are critical for adjusting the frequency of this behavior. We have added these new data to the revised manuscript (Fig. 5a).

8. While Fig. 5 a-c show that the mGLUR mutant *mgl-2(tm355)* is able suppress the effect of *glt-1*, this effect seems to be partial. Either the mGLuR mutant is not a null, has some perdurance (I am not sure about the crosses here) or alternatively the effect of *glt-1* on repetitive reversal behavior is not solely dependent on mGLuR but might also act via some other mechanism. This should be discussed by the authors.

This is an important point. The *mgl-2(tm355)* deletion allele is predicted to be *null*, as it generates a premature stop codon, leading to a truncated protein lacking several transmembrane domains. Thus it is likely that alternative pathway(s) also affect repetitive behavior. We added this discussion point to the revised manuscript.

Reviewer #2 (Remarks to the Author):

In this manuscript, Katz et al report of a simple feedback process that can explain repetitive behavior in *C. elegans* worms. The underlying mechanism involves inactivation of the glutamate transporter GLT-1 in CEPsh cells which results in glutamate spillover. The un-cleared glutamate, secreted by the RIM neuron, presumably rebinds the glutamate receptor MGL-2, expressed on RIM, to initiate another activation cycle. Postsynaptic to RIM is the command neuron AVA that is repeatedly activated to promote a repetitive reversal behavior.

The authors performed a comprehensive rigorous work and meticulously analyzed the data applying the appropriate statistical tests. I am less certain about the novelty of the drawn conclusions as impaired glutamate signaling has already been established as a possible cause of various behavioral disorders, including repetitive behaviors in humans, such as OCD (see refs 34-35). Nevertheless, I find the simple autocrine mechanism, revealed in *C. elegans* worms, an intriguing and elegant demonstration of how aberrant glutamate clearance can lead to repetitive behaviors.

We thank the reviewer for their positive view of our manuscript. We would also like to add that besides the mechanistic understanding of the synaptic processes controlling repetitive behavior, which is the major novel finding of our manuscript, our studies also establish *C. elegans* as an outstanding model system for understanding fundamental aspects of astrocyte biology, and provide the first invertebrate platform for assaying repetitive behavior. This opens the field to performing high-throughput analyses of this aberrant behavior in a way that had not been previously possible in the mouse.

Specific comments for the authors:

Major:

1. The behavioral assays that score reversal rates (e.g. figs 2-3,5) are performed on plates with no food. Specifically, scoring starts 20 min. after the worms were placed on the plate. This is also the typical time in which worms switch from local food search (characterized by many reversals) to dispersal (long runs and less reversals). It is possible that defective glutamate signaling affects worms' sense of starvation, and consequently, the timing of this behavioral switch which might affect reversal rates when compared to WT animals. To exclude the possibility that starvation processes bias the reversal rates, the authors ought to repeat these experiments on plates with food. At least to verify that *glt-1* mutants still show enhanced reversal rates compared to WT and that GLT-1 rescue in CEPsh glia is similar to WT. In such experiments, scoring should start 2-3 after the transfer of the animals to the assay plate. Such conditions will also match the conditions used for functional analyses of Ca and Glu release (fig 4).

The reviewer raises an interesting and important point. Indeed, we intentionally performed our behavior studies during the dispersal phase discussed by the reviewer, as reversals are infrequent during this phase, increasing our sensitivity to changes in reversal rates. We now emphasize this point in the revised manuscript.

We could not perform the exact study requested by the reviewer, precisely because animal movement patterns on food are very different from those off food. For example, animals have many reversal events on food, but these are of short duration, rarely extending beyond a single body bend, and are therefore qualitatively different from those off food. Therefore, we cannot be confident that we are comparing the same behaviors on and off food. To circumvent this issue we performed the following studies:

1. To confirm that the effects of *glt-1* are not related to the starvation state of the animals, we studied the effects of *glt-1* loss on reversals in a different behavioral paradigm, reversal following mechanical prodding of the animal's head (Fig. 3f). These experiments were done on plates covered with a thin bacterial-lawn (see methods), and animals were, therefore, not starved. Since *glt-1* mutants show induced repetitive reversals in both behavioral assays, it is unlikely that *glt-1* functions to control the sense of starvation. We have modified the manuscript to emphasize this result more clearly.

2. If GLT-1 is a satiety sensor, we would expect *glt-1* mutants to show similar behavior in the presence or absence of food. We tested this directly by measuring a number of locomotion parameters, including speed, dwelling/pausing, the fraction of reversals that terminate with omega turns, reversal rates, and repetitive reversals. As seen in the figure below, *glt-1* mutants differ dramatically in behavior on and off food, with similar responses to food removal as wild-type animals. This suggests that *glt-1* animals have no problem detecting and responding to food cues. Furthermore, the only behaviors off food that show differences from the wild type are overall reversal rate and repetitive reversals, suggesting that *glt-1* has very specific effects, that are unlikely to be explained by a general defect in satiety detection or response.

2. In line 184, the authors state that: “Wild-type animals initially show no glutamate release or calcium induction events in our assay. However, after 4 min ± 24 sec of habituation to the food cue, on average, spontaneous glutamate release events onto AVA ensue”. Looking at suppl. fig

4, it shows that this is not correct and at least half of the animals do show activity in the first 4.5 minutes. Actually, based on the suppl. figure, contrasting WT (fig 4a) and *glt-1* (fig 4b) is misleading as these are not the typical representative traces.

We are a bit confused by the first part of this comment. The response time of wild type (or *glt-1* mutants) is roughly normally distributed (see figure to the right). In a normal distribution, one would expect half of events to precede the average response time (i.e. 4 minutes), and half to occur after. This is precisely what we see: “4 min ± 24 sec” indicates the mean ± the standard error.

Regarding the second point, the traces presented in figure 4a-f aim to illustrate the differences in oscillations between wild type and *glt-1* mutants, and to help the reader follow our frequency analysis (as wavelet analysis is not routinely used). Indeed, given the variability among traces (which we discuss in the manuscript and which necessitated wavelet analysis), it is somewhat difficult to define a “representative trace”. Nonetheless, to address the reviewer’s concern, and to help the readers appreciate the differences in response kinetics, we have now added to the revised manuscript a dozen traces (including their wavelet plots) for each genotype (Supplementary Fig. 5c,d), to show the range of effects seen.

3. In fig. 5a, the repetitive behavior index of *glt-1(ok206)* is nearly 1.2 while in fig 3d the value is 0.7. I guess these values come from two independent experimental sets, but the difference seems a bit high. For example, if the value was 0.7 in fig 5a then it would not be significantly different from the double mutant *glt-1;mgl-2* as shown in fig 5a. It is particularly important to verify this issue since it is shown that the reversal rates of *glt-1* and *glt-1;mgl-2* are similar (suppl. fig 6a).

This is an important comment that brings up a general issue in all animal behavior assays. It is an empirical fact that animal behavior assays are highly variable. For example, some behavioral studies in *C. elegans* can only be performed seasonally, as summer humidity likely changes many behavioral parameters. In general, from our own experience and that of the field, we only compare experiments done over a period of several weeks. We also run a complete set of controls on each day of data collection, as historical controls are unreliable. This allows us to compare trends in experiments performed on different date intervals.

To illustrate our approach, the top graph on the right shows all repetitive-behavior-index (RI) values for wild type and *glt-1* mutants for all movies used in the manuscript, and are from experiments done on different date intervals. Values derived from a single experiment are labeled by the same color. As can be seen, the differences in RI between the two strains for any specific experiment are clear. Yet, different experiments are variable. That this variability is normally distributed (bottom graph) suggests that the data obeys the Central Limit Theorem, suggesting similar data distribution for each experiment.

Therefore, while there are differences in the precise values of the repetitive behavior index between different studies, the same trends are seen across

studies, and we believe that this is the relevant parameter to focus on.

4. On a related note, in fig 5a RI of *glt-1* is significantly higher than *glt-1;mgl-2*. However, in fig 5b, if I read correctly the notations above the bars, then the RIs of these two strains are not significantly different. How come?

The differences between these two experiments arise from the sensitivity of the ANOVA statistical platform to the variance among and between groups. Direct comparison of the RI values between *glt-1* and *glt-1; mgl-2* double mutants using a *t*-test indicates significant difference in both experiments. To address this issue, we repeated this experiment, increasing the number of replicates, allowing significance to be shown in both figures. As all recordings were done at the same time, the original panels (a and b) are now combined and presented in the same graph (Fig. 5a).

5. When RIM was light activated, the authors scored quick reversals during the short light stimulus (5 seconds as noted in methods line 683). This is very different from the oscillatory responses and repetitive behavior described in all other experiments, which were typically ~25 seconds. What is the reason that the scoring was done differently? According to the suggested model, I would expect to find the 25s period of oscillations following a single light activation of RIM. The authors should assay that in order to support their claims and the suggested model where the feedback through RIM underlies the observed repetitive phenotypes.

We agree with the reviewer that scoring the entire temporal distribution of reversal events, predicted to peak at 25 sec, in the optogenetic studies would be more consistent with the other behavior assays in the manuscript. Nonetheless, there is an important technical obstacle that forced us to use an alternative method of quantification here, focusing on only an early time point.

Specifically, in the optogenetic studies, the reversal event distribution is a convolution of the spontaneous reversal distribution and that evoked by light stimulation. At short intervals after light stimulation, spontaneous events are largely negligible, and can be subtracted from the total signal by scoring animals exposed to light but not to retinal, which transduces light stimulation to effect ion channel opening. This allows us to obtain a measure of repetitive behavior induced by light stimulation. At longer intervals, the likelihood of spontaneous reversal events increases dramatically, peaking at ~25 sec. Subtracting the observed reversal rate from the spontaneous rate obtained from retinal(-) controls yields a very noisy distribution, as these rates are of similar order of magnitude. Thus, it is essentially impossible to tell whether an event occurred stochastically, or was a response to the light stimulation.

Therefore, instead of scoring the entire reversal distribution, as we did for spontaneous reversals, we opted only to look at short times after optogenetic stimulation.

6. The authors show that RI is negligible in *egl-30/Gaq;glt-1* mutants but significantly increases in *goa-1/Gao;glt-1* mutants (as well as glutamate secretion and AVA activation events). However, these genes are widely expressed in the nervous system and many neurons could collectively contribute to the observed phenotypes. Thus, unless the genes are introduced to rescue in a neuron-specific manner it is practically impossible to tell whether RI and AVA activations are directly affected by MGL-2-mediated Ga signaling in the RIM neuron.

This is an excellent point. At the reviewer's suggestion we expressed EGL-30 specifically in RIM and assessed rescue. As shown in the new Fig. 6b, repetitive reversals are significantly enhanced in *egl-30(ad806); glt-1(ok206)* double mutants, supporting the idea that EGL-30 acts, at least in part, in RIM. Expression of EGL-30 in RIM in *egl-30* single mutants had no effect on repetitive reversal rate.

Furthermore, as can be seen in Supplementary Fig. 8c, the overall reversal rate (not repetitive reversals, just the average rate) of *egl-30* mutants is significantly lower than that of wild type, and

EGL-30 expression in RIM has no effect on this rate, suggesting that EGL-30 likely acts in RIM to control repetitive behavior, and elsewhere to control overall reversal rate. Thus, although reversals occur infrequently in *egl-30; glt-1* double mutants, when they do happen, and *egl-30* is restored only to RIM, a repetitive bout will ensue.

Because of the low reversal rate in *egl-30; glt-1*; RIM::*egl-30* animals, the probability of catching an animal exhibiting AVA activation in our recordings is very low, and we were not able to do so, given the low throughput of this method. We therefore used another approach.

Specifically, we predicted that *egl-30* gain of function single mutants should exhibit spontaneous repetitive oscillations of AVA. Consistent with our model, we observed such oscillations comparable to those of single *glt-1(-)* mutants (see heatmaps below).

Taken together, our data are consistent with a model in which *egl-30* functions, at least in part, in RIM to promote repetitive behavior.

7. Ca and Glu traces are oscillatory and highly correlative. However, if the proposed model is that Glu is not cleared from synaptic and extra-synaptic sites, should not glutamate levels remain high, or possibly, only mildly decrease? It seems that glutamate concentrations always return to basal levels (fig 4b).

We do not propose that glutamate is not cleared and remains continually high in synapses of *glt-1* mutants. Rather, the key to our model is that there is only transient increase in glutamate over normal levels following synaptic release, and that glutamate is eventually cleared by means other than GLT-1-dependent uptake (diffusion, for example is estimated to occur rapidly at $\sim 0.4 \mu\text{m}^2/\text{msec}^3$). This transient increase induces non-evoked glutamate release through MGL-2, resulting in another round of glutamate release, and so on.

Like the reviewer, we were also initially surprised to see the glutamate oscillations. However, closer examination of the literature revealed that diffusion is a significant factor in the clearance of synaptic glutamate³, especially at synapses that are not fully ensheathed by glial process, like *C. elegans* synapses. Thus, even in the absence of active uptake, glutamate is not expected to remain at the synaptic cleft for a prolonged period of time.

Please also see our detailed response to Reviewer #1 point #5 addressing a similar concern.

8. Related to the above, could the authors explain why spillover-based oscillatory periods are ~ 25 seconds long? Naively, I would expect that following RIM activation, glutamate will remain in excess at synapses and extra-synaptic regions and instantly induce MGL-2 mediated RIM reactivation (similar to the quick reversals observed following light-activation of RIM).

This is a very good point. We believe that the delay in re-release may be a result of the slow kinetics of MGL-2 transduction following glutamate engagement. While ionotropic receptors, such as the AMPA receptor, transduce glutamate binding quickly, as they are ion channels

themselves, MGL-2 is a metabotropic glutamate receptor, which indirectly leads to opening of other membrane ion channels, with transduction kinetics of seconds up to tens of seconds⁴. Thus, while synaptic glutamate, allowed to diffuse in the absence of GLT-1, would rapidly trigger MGL-2, the ensuing signal transduction would be rate limiting, promoting unevoked glutamate release with a time scale of seconds to tens of seconds.

We added a discussion of this point to the revised manuscript.

Minor:

- Supplementary Fig. 1e: It is not clear what each of the three CEPsh-glia-abl bars denotes. Also, why reversal rate values of WT and ablated animals are markedly different between suppl fig 1e and fig 2a (~two-fold ratio)?

These are three independent integrated lines that were used to verify that the behavioral defects are not caused by mutations that may occur during the genomic integration of the plasmid. We now clarify this in the figure legend.

- Supplementary Fig. 3 legend: typo in 'intersept'.

Corrected, thank you.

Reviewer #3 (Remarks to the Author):

The authors use *C. elegans* to probe the role of glutamate signaling, particularly glutamate spillover at the RIM::AVA synapse in affecting animal behavior. They identify a conserved glial glutamate transporter, *glt-1* that clears glutamate at this synapse. Specifically, they show that GLT-1 mutants fail to clear this synapse, causing activation of *mgl-2* on the RIM axon. This forms a positive feedback loop, leading to repetitive reversals. I do have a couple of concerns that would improve their manuscript (listed in no particular order).

Major concerns

1. It is possible that MGL-2 might also act in RIM dendrites. In this case, the interesting synapse is between RIM::neuron (not AVA). Further, the altered RIM activity then modifies AVA activity through traditional neurotransmission.

This is an important point that we think needs to be clarified.

Unlike many mammalian neurons, *C. elegans* interneurons are mostly monopolar, with no obvious anatomical segregation of postsynaptic and presynaptic sites. Thus, input and output synapses can be found at close proximity. Therefore, at the level of fluorescence imaging, we cannot determine whether MGL-2 localizes pre or post-synaptically (although it is clear MGL-2 is not directly at the presynaptic site).

However, the experiment presented in Fig. 5c of the manuscript shows that RIM can sense spillover of glutamate released cell-autonomously from RIM itself through optogenetic stimulation. Thus, oscillations in AVA in this case are not initiated at RIM input synapses.

Nonetheless, in the context of repetitive behavior observed in *glt-1* mutants, we think the reviewer is correct that other synapses could also experience similar positive feedback we observe at the RIM::AVA synapse, and that the effects on AVA would then be a result of signal propagation downstream of a higher-level synapse.

We have added this point of discussion to the revised manuscript.

2. Is glial *glt-1* really important? It looks like neuronal *glt-1* might also have a role in affecting neural activity and behavior (Fig. 2c). Perhaps tissue selective knockdowns could be used to sort out the role of the two transporters.

This is an important point that we would like to clarify. As we discuss in our original manuscript, analysis of expression of a *glt-1* promoter::GFP transgene reveals only strong expression in CEPsh glia and weaker expression in head muscles. No expression is detected in neurons. This is corroborated by our RNA sequencing data, which shows a large enrichment of GLT-1 expression in glia over non-glial cells (many of which are neurons and muscles).

Regardless of the expression site, the main point we are trying to make is that *glt-1* affects extracellular levels/dynamics of glutamate released from synapses. To test this hypothesis, we sought an artificial way to express GLT-1 at glutamatergic synapses by ectopic expression in neurons. This indeed shows the most efficient rescue, suggesting that synaptic proximity likely plays a role in the effects of GLT-1 (Fig. 2c).

We have now modified the text to make these ideas more clear.

3. Statistics. It looks like the authors are assuming normality for all statistical measures. It is probably fine for most of the data, except for reorientations (a naturally bounded metric). This is particularly problematic when comparing genotypes with small numbers of reorientations.

We are not exactly sure what the reviewer means by “reorientations”, and why this is a bounded quantity. Perhaps the reviewer is referencing the main parameter we study in the paper, reversal behavior, although since the time spent reversing, even in *glt-1* mutants, is a small percentage of total movement, the maximal possible reversal rate is never approached, and the statistics can therefore be considered approximately normal.

Regardless of this issue, the repetitive behavior index (RI) for each strain is obtained from multiple identically independent experiments. By the Central Limit Theorem, the means of these experiments will therefore be normally distributed (see graph). Thus, comparisons between strains are comparing normally distributed data.

4. The authors claim that glutamatergic synapses are close to glial GLT-1, but this is not clear (Figure 2b). Perhaps, they could add an additional synapse to their analysis.

As suggested by the reviewer, to better illustrate the close proximity between CEPsh-expressed GLT-1 and glutamatergic synapses, we added individual images from a z-stack series of optical sections taken at ~1.5 μm intervals through the neuropil (Supplementary Fig 3). We also replaced the original image of Fig. 2b with one of these optical sections. This slice series demonstrates clearly that every glutamatergic synapse is close to a glial process (<0.5 μm , and probably <0.2 μm , based on the limit of object resolution on this microscope).

5. It looks like the wavelet analysis shown in Fig. 4cdef wherein *glt-1* mutants have higher frequency oscillations at various time points seem to be derived from 1 animal? If so, I would recommend adding a few more to the analysis.

As suggested by the reviewer, we have now added to the revised manuscript a dozen calcium traces and wavelet plots for wild type and for *glt-1* mutants (Supplementary Fig. 5c, d).

Nonetheless, we would like to clarify that the traces presented in Fig. 4a-f are only used to illustrate the overall phenomenon and to explain how these activity frequencies were analyzed

(as wavelet analysis is not a commonly used method). All of the conclusions in our manuscript are derived from analysis of the recordings of many individual animals for each strain using the appropriate statistical tests (in particular, see Fig. 4g, h). Heatmaps representing all the traces that were analyzed in each experiment are included in the Supplementary Data.

6. The authors argue that MGL-2 function in RIM. I would recommend knocking down MGL-2 in RIM specifically to test their hypothesis. It is possible that MGL-2 acts in additional neurons to affect AVA function.

This is another important point that needs to be clarified. Indeed, we focus on MGL-2 function in RIM as a model to study the molecular mechanisms by which glutamate spillover drives repetitive behavior. However, we don't argue that repetitive behavior is only induced by RIM. On the contrary, it is likely that CEPsh glia have a general role in control of glutamate signaling, and that any glutamatergic neuron also expressing MGL-2 can act as an oscillation generator upon glutamate spillover.

To directly illustrate this point, we now present data showing that MGL-2 expression in another neuron, AIB, is also sufficient to significantly increase repetitive reversals in *mgl-2*; *glt-1* double mutants (Fig. 5b), and we modified the text to better emphasize this point.

Minor Concerns

1. Figure 1b. Y-axis is confusing. Maybe there is a better way to present this data.

This is the standard output of the clustering code, a dendrogram illustrating the relationships between individual genes. We modified the figure legend to better explain what the y-axis represents.

2. Figure 3c is a bit confusing. I thought that the plot shown here was the observed-exponential fit?

The plot represents the differences in expected-observed reversals probabilities only for reversals that occur in time intervals up to 65 seconds. It does not include reversals that occur at long time intervals (which by definition are not repetitive). Indeed, applying this analysis at long time intervals would result in negative values (see the last two time points of *glt-1* mutants in Fig. 3c).

3. Figure 4i feels out of order. I would suggest that animal movement is correlated with calcium transients.

It is well established in the literature that AVA activation is associated with reversal response. The question that we aimed to address is whether the high frequency of activity oscillations in AVA can be translated into a behavioral response. For that reason we present this graph following the description of the oscillatory activity.

4. Statistics in Figure 5c.

We are not sure to what the reviewer is referring, but perhaps the issue being raised is the observation that in *mgl-2*; *glt-1* double mutants the reduction in repetitive reversals does not show statistical significance. As indicated in Fig. 5a,b, when scored over the whole range of recorded reversals events, mutations in both *mgl-2* and *glt-1* result in partial (yet significant) rescue of the *glt-1(-)*-increased repetitive reversals. During the optogenetic activation of RIM, reversals are scored over a short time period, limiting the total number of events (please see our response to Reviewer #2 comment #5 for more details). Together, these two limitations may affect the strength of the statistical testing in the double mutants. However, while not significant at the level of $p < 0.05$, the trend towards fewer repetitive events in the double mutant animals is consistent with our observations in Fig. 5a,b.

We have now modified the text and the figure legends to clarify this point.

- 1 Kawano, T., Takuwa, K. & Nakajima, T. Structure and activity of a new form of the glutamate transporter of the nematode *Caenorhabditis elegans*. *Biosci Biotechnol Biochem* **61**, 927-929, (1997).
- 2 Rothstein, J. D. *et al.* Knockout of glutamate transporters reveals a major role for astroglial transport in excitotoxicity and clearance of glutamate. *Neuron* **16**, 675-686, (1996).
- 3 Kessler, J. P. Control of cleft glutamate concentration and glutamate spill-out by perisynaptic glia: uptake and diffusion barriers. *PLoS One* **8**, e70791, (2013).
- 4 Abe, T. *et al.* Molecular characterization of a novel metabotropic glutamate receptor mGluR5 coupled to inositol phosphate/Ca²⁺ signal transduction. *J. Biol. Chem.* **267**, 13361-13368, (1992).

Reviewers' Comments:

Reviewer #1:

Remarks to the Author:

The authors have done a great job addressing all of my concerns, as well as other reviewers. This is a very nice study.

Reviewer #2:

Remarks to the Author:

The authors have satisfactorily addressed all my concerns, and I have no further questions.

Reviewer #3:

Remarks to the Author:

I was most concerned about two points.

1. Is glt-1 activity crucial pre- or post-synaptic to RIM?

I find the explanation and their RIM optogenetics experiment to be pretty convincing. glt-1 compounding the effect of RIM activation (+ the noted effects on downstream AVA) agrees with their model of poor clearance of RIM-released glutamate causing repetitive reversals.

2. Is glt-1 expression in glia actually important?

Their expression data showing glt-1 is really only expressed in glia is key here (I must have missed this to begin with). I buy their explanation that neuronal rescues are successful due to proximity.

Their statistics seem fine to me as well.

We are happy to see that we addressed all of the reviewers concerns and that they find the revised manuscript suitable for publication.

REVIEWERS' COMMENTS:

Reviewer #1 (Remarks to the Author):

The authors have done a great job addressing all of my concerns, as well as other reviewers. This is a very nice study.

Reviewer #2 (Remarks to the Author):

The authors have satisfactorily addressed all my concerns, and I have no further questions.

Reviewer #3 (Remarks to the Author):

I was most concerned about two points.

1. Is glt-1 activity crucial pre- or post-synaptic to RIM?

I find the explanation and their RIM optogenetics experiment to be pretty convincing. glt-1 compounding the effect of RIM activation (+ the noted effects on downstream AVA) agrees with their model of poor clearance of RIM-released glutamate causing repetitive reversals.

2. Is glt-1 expression in glia actually important?

Their expression data showing glt-1 is really only expressed in glia is key here (I must have missed this to begin with). I buy their explanation that neuronal rescues are successful due to proximity.

Their statistics seem fine to me as well.